# `InnoEval`: On Research Idea Evaluation as a Knowledge-Grounded, Multi-Perspective Reasoning Problem

**Shuofei Qiao** [1 2]  **Yunxiang Wei** [1]  **Xuehai Wang** [1]  **Bin Wu** [2]  **Boyang Xue** [3]  **Ningyu Zhang** [1]  **Hossein A. Rahmani** [2]
**Yanshan Wang** [1]  **Qiang Zhang** [1]  **Keyan Ding** [1]  **Jeff Z. Pan** [4]  **Huajun Chen** [1]  **Emine Yilmaz** [2]

zjunlp/InnoEval    innoeval.zjukg.cn

## Abstract

The rapid evolution of Large Language Models has catalyzed a surge in scientific idea production, yet this leap has not been accompanied by a matching advance in idea evaluation. The fundamental nature of scientific evaluation needs knowledgeable grounding, collective deliberation, and multi-criteria decision-making. However, existing idea evaluation methods often suffer from narrow knowledge horizons, flattened evaluation dimensions, and the inherent bias in LLM-as-a-Judge. To address these, we regard idea evaluation as a knowledge-grounded, multi-perspective reasoning problem and introduce `InnoEval`, a deep innovation evaluation framework designed to emulate human-level idea assessment. We apply a heterogeneous deep knowledge search engine that retrieves and grounds dynamic evidence from diverse online sources. We further achieve review consensus with an innovation review board containing reviewers with distinct academic backgrounds, enabling a multi-dimensional decoupled evaluation across multiple metrics. We construct comprehensive datasets derived from authoritative peer-reviewed submissions to benchmark `InnoEval`. Experiments demonstrate that `InnoEval` can consistently outperform baselines in point-wise, pair-wise, and group-wise evaluation tasks, exhibiting judgment patterns and consensus highly aligned with human experts.

[1]Zhejiang University [2]University College London [3]The Chinese University of Hong Kong [4]The University of Edinburgh. Correspondence to: Ningyu Zhang, Huajun Chen. Contact: Shuofei Qiao <shuofei@zju.edu.cn>, Ningyu Zhang <zhangningyu@zju.edu.cn>, Huajun Chen <huajunsir@zju.edu.cn>, Emine Yilmaz <emine.yilmaz@ucl.ac.uk>.

*Proceedings of the $43^{rd}$ International Conference on Machine Learning*, Seoul, South Korea. PMLR 306, 2026. Copyright 2026 by the author(s).

## 1. Introduction

The integration of Large Language Models (LLMs) into scientific discovery (Chen et al., 2025c; Schmidgall et al., 2025; Lu et al., 2024; Chai et al., 2025; Tang et al., 2025) has catalyzed a paradigm shift, enabling the generation of research ideas at an unprecedented scale (Si et al., 2025; Baek et al., 2025; Li et al., 2024; Novikov et al., 2025). However, this "generative explosion" has outpaced our evaluative capability, creating a critical bottleneck at the very outset of the research pipeline. Current innovation evaluation remains heavily dependent on scarce and highly specialized human experts, which is not only time-consuming and costly, but its inherent subjectivity and limited scope also risk overlooking potentially high-value ideas. Consequently, developing an automated, systematic, and yet human-expert-level idea evaluation framework has become an urgent necessity.

To address this, we must first revisit the fundamental nature of scientific evaluation. Ideally, evaluating a research idea is not a static generation task, but a holistic epistemic verification process, which can be intrinsically defined by three cardinal principles: **(i) Knowledgeable Grounding** (Quintane et al., 2011; Du Plessis, 2007), where idea is a knowledge-intensive entity cross-referenced against the entire living ecosystem of theory and practice; **(ii) Collective Deliberation** (Kameda et al., 2022; Navajas et al., 2018), where the quality of assessment emerges from the fusion of diverse expert perspectives, not a single authority; **(iii) Multi-criteria Decision Making** (Majumder, 2015), where an idea's inherent complexity is honored through the union of multifaceted attributes (e.g., novelty, feasibility, impact) rather than the compression into one or two dimensions.

However, achieving the above principles confronts several challenges that together constitute the core gap between current automated idea evaluation tools (Moussa et al., 2025; Zhang et al., 2026; Agarwal et al., 2025; Feng et al., 2025; Shahid et al., 2025) and an ideal evaluation system. First is the **(i)** *narrowness of knowledge horizon*. Existing methods are largely confined to static academic papers, overlooking the complete ecosystem of research innovation. This neglect of "living knowledge" renders evaluations divorced from

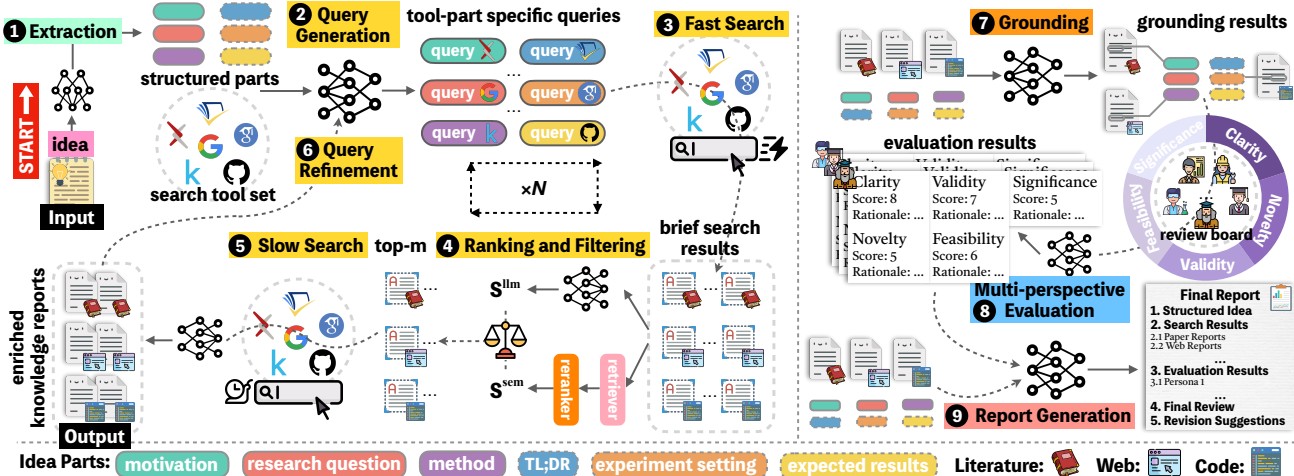

*Figure 1.* **Framework of InnoEval**. Structured idea parts with dashed boxes are optional. Given the raw-text idea, the deep knowledge search engine (left-hand) iterates $N$ times, ultimately yielding enriched knowledge reports categorized into three types: literature, web, and code. During evaluation, each metric is assessed by a dedicated evaluator agent, and users can freely register additional metrics.

practical realities. Second is the **(ii)** *overlook of review consensus*. Mainstream approaches directly using LLM-as-a-Judge fossilize the models' inherent biases into de facto evaluation criteria, failing to emulate the deliberation among distinct perspectives. Last is the **(iii)** *flattening of evaluation dimensions*. Existing methods typically obscure the independence and internal tensions among multiple attributes embedded in a complete research idea, fail to deliver useful feedback, and violate the multi-criteria decision-making that systematic scientific evaluation should adhere to.

In response to these challenges, we regard idea evaluation as a knowledge-grounded, multi-perspective reasoning problem and introduce **InnoEval**, a deep innovation evaluation framework that emulates the full scholarly review process performed by human experts. To counter the narrowness of knowledge horizons, we design a **(i)** *heterogeneous deep knowledge search engine*, which can concurrently obtain living knowledge from online literature, web opinions, and code repos, combining fast search with slow reading for both effectiveness and efficiency. Through multi-round query refinement and hybrid scoring that blends semantic similarity with LLM judgment, it filters highly relevant, high-quality background knowledge from diverse sources, thereby constructing a living, comprehensive, and cross-domain evidential knowledge base for grounding and evaluation. Next, to achieve review consensus, we simulate an **(ii)** *innovation review board* to perform multi-perspective evaluation by sampling multiple reviewers with distinct backgrounds, expertise, and preferences from a carefully curated academic persona pool. We partially mask the searched knowledge according to each reviewer's role to simulate real human cognition. Each reviewer will independently judge the research idea, thereby mitigating the bias inherent in a single judge. During actual evaluation, to overcome flattened evaluation dimensions, we perform a **(iii)**

*multi-dimensional decoupled evaluation*, supporting customizable multi-criteria assessment initialized across five dimensions: *Clarity*, *Novelty*, *Feasibility*, *Validity*, and *Significance*. Each dimension is assessed by a specialized evaluator agent that conducts in-depth analyses grounded in both retrieved heterogeneous knowledge and its own internal knowledge. All multi-dimensional evaluations from the personified reviewers are aggregated into an actionable report that provides cited knowledge evidence, structured evaluation analyses and scores, a meta-review with specific decisions, and concrete directions for further improvement.

To evaluate InnoEval, we extract real-world ideas from authoritative peer-reviewed papers and construct datasets that respectively inspect single-idea evaluation, pairwise-idea comparison, and group-idea ranking capabilities. Quantitatively, InnoEval outperforms the strongest baseline by 16.18% F1 score in three-class point-wise prediction, by roughly 5% accuracy in pair-wise comparison, and by 7.56% accuracy in group-wise ranking. Qualitatively, evaluation reports produced by InnoEval achieve a win rate exceeding 70% against all baselines in *Overall Quality*. In human evaluation, InnoEval's scores exhibit high correlation with human judgments across all metrics. Further ablation studies confirm the effectiveness of each of our proposed modules. Exploratory analyses reveal that InnoEval's evaluation exhibits patterns and phenomena aligned well with human cognition, indicating that it can faithfully simulate the real process of human innovation evaluation.

## 2. Problem Definition

The idea in our paper can be expressed in any format, ranging from a preliminary hypothesis to a fully-fledged idea presented in a mature paper. Regardless of its form, we formally define an idea $\mathcal{I}$ as a structured six-tuple $\mathcal{I} = ($**TLDR**,

**Motis**, **ResQues**, **Meths**, **ExpSets**, **ExpRes**), respectively representing a short summary of the idea, motivations, research questions, methods, experimental settings, and expected results. Here **TLDR**, **ExpSets**, and **ExpRes** are optional based on the idea's maturity. Due to the time-sensitive nature of research ideas, we also equip each idea with a timestamp $t$, denoting the temporal standpoint from which the evaluation should be conducted. $t$ will default to the latest time if not explicitly specified. Given an evaluation system $\mathcal{F}$, our goal is to obtain a final evaluation report $P$, which can be categorized into two types.

**Point-wise.** Evaluate a single idea $\mathcal{I}$ and generate an evaluation report $P_{\text{point}}$ that contains background knowledge $\mathcal{K}$ of the idea searched by the system, provides multi-dimensional assessment results $E_{\text{point}}$, and offers concrete revision suggestions $\mathcal{V}$ for future improvements of the idea:

$$P_{\text{point}} = \{\mathcal{K}, \mathcal{V}, E_{\text{point}}\} = \mathcal{F}(\mathcal{I}) \tag{1}$$

**Group-wise.** Rank a group of $n(n \geq 2)$ ideas $\{\mathcal{I}_i\}_{i=1}^n$ on the same topic and deliver individual reports for each idea together with the final ranking evaluation and results $E_{\text{group}}$:

$$P_{\text{group}} = \{\{P_{\text{point}}^{\mathcal{I}_i}\}_{i=1}^n, E_{\text{group}}\} = \mathcal{F}(\{\mathcal{I}_i\}_{i=1}^n) \tag{2}$$

To facilitate quantitative evaluation of the system, we also include measurable outcomes for each setting. For point-wise, $E_{\text{point}}$ also contains a final decision $d_{\text{point}} \in \{$**Reject**, **Poster**, **Spotlight**, **Oral**$\}$ of the idea. For group-wise setting, $E_{\text{group}}$ provides a complete ranked list $d_{\text{group}}$ of all ideas.

## 3. InnoEval

**Framework Overview.** As shown in Fig.1, InnoEval begins by extracting structured parts from the raw-text idea, followed by a heterogeneous deep-knowledge search engine that iteratively acquires and filters high-quality background knowledge from diverse online sources. The retrieved results, after fine-grained grounding, serve as key references for multi-dimensional and multi-perspective evaluation, empowered by a group of dimension-specialized evaluator agents and an innovation review board composed of diverse academic personas. Finally, an actionable evaluation report will be synthesized, including comprehensive meta-reviews and concrete suggestions for future revision. All the notations in this section are summarized in Tab.3.

### 3.1. Heterogeneous Deep Knowledge Search

To maximize usability, we impose no constraints on the format of an idea to be evaluated. However, to enable precise and efficient knowledge searching and grounding, we first distill the raw textual idea $T$ into the structured representation defined in §2. This process is carried out by an

Extraction Agent $\mathcal{M}_e$, formally defined as $\mathcal{I} = \mathcal{M}_e(T)$. Then, an exhaustive **Search Agent** $\mathcal{M}_s$ is deployed to systematically retrieve academic resources relevant to the idea. To ensure *comprehensiveness*, we expose our research agent to heterogeneous online sources spanning scholarly literature ($\times$ arXiv, $\triangledown$ Semantic Scholar, $\circledS$ Google Scholar), web content ($G$ Google Search), and open-source code repositories ($\bigcirc$ Github, $k$ Kaggle). By relying solely on online sources, we avoid the staleness inherent to static local corpora and retain *flexibility* to incorporate emerging knowledge. Moreover, timestamps can be effortlessly integrated into online searching APIs, allowing us to explicitly account for the *timeliness*, a critical attribute of all research ideas. To ensure retrieval *quality*, we employ an iterative search strategy that simultaneously scales breadth and drills for depth. We further integrate a fast-slow search hybrid into the deep iterative process to improve search *efficiency*.

**Fast Knowledge Search.** Concretely, for every component $p \in \mathcal{I}$ and every search tool $u \in (\times, \triangledown, \circledS, G, \bigcirc, k)$, the search agent first generates a group of tailored queries and then enriches them with synonym expansions, an essential step given that equivalent concepts are routinely phrased differently across communities, yielding the final query set $\mathcal{Q}_{p,u} = \mathcal{M}_s(p, u)$. We then directly use the corresponding API for fast retrieval to obtain brief search results:

$$\widetilde{\mathcal{K}}_{p,u} = u(\mathcal{Q}_{p,u}, t). \tag{3}$$

We partition all the briefly searched knowledge $\widetilde{\mathcal{K}}$ into three types. Literature knowledge $\widetilde{\mathcal{K}}_l$ contains papers' meta-info such as title, abstract, venue, etc. Web $\widetilde{\mathcal{K}}_w$ and code knowledge $\widetilde{\mathcal{K}}_c$ include the snapshots of the webpages or repos. We introduce a **timestamp $t$** during search to separately retrieve and process knowledge published before and after $t$. Pre-knowledge is used to evaluate the idea, whereas post-knowldege are used to suggest revisions. **But for notational simplicity, we do not distinguish them symbolically**.

**Knowledge Ranking and Filtering.** After discarding unusable knowledge with missing information or inaccessible links, we rank all remaining knowledge using a hybrid scoring function. Then we calculate the semantic similarity between the idea $\mathcal{I}$ and each piece of knowledge with an embedding model and retain the top $3m$ items for every knowledge type. A reranking model refines these $3m$ candidates and produces the reranked scores $\mathcal{S}^{\text{sem}}$. We condense the above procedure into the following formulation:

$$\mathcal{S}_j^{\text{sem}} = \texttt{Rerank}(\texttt{Embed}(\mathcal{I}, \widetilde{\mathcal{K}}_j)), j \in \{l, w, c\}. \tag{4}$$

We then use $\mathcal{M}_s$ as a judge model to assess the relevance and quality of each knowledge, incorporating factors such as paper citations, published venue, website popularity, repo stars, etc., and obtain the model-as-judge scores:

$$\mathcal{S}_j^{\text{llm}} = \mathcal{M}_s(\mathcal{I}, \widetilde{\mathcal{K}}_j), \quad j \in \{l, w, c\}. \tag{5}$$

Finally, we re-weight the two scores with a coefficient $\alpha$ and select the top-$m$ knowledge pieces for each category $j$:

$$\widetilde{\mathcal{K}}_j^* = \texttt{Top}_m(\widetilde{\mathcal{K}}_j, \mathcal{S}_j), \ \mathcal{S}_j = \alpha \mathcal{S}_j^{\text{sem}} + (1-\alpha)\mathcal{S}_j^{\text{llm}}. \quad (6)$$

We adopt a hybrid scoring function to simultaneously mitigate the fragility of semantic similarity, and the bias and hallucination inherent in model-as-judge evaluation.

**Slow Knowledge Search.** After obtaining the carefully filtered knowledge, we apply a slow search to enrich their content. For literature-type, we retrieve the full PDF, convert it into raw text, and process it with $\mathcal{M}_e$ to yield structured text that serves as the enriched knowledge $\mathcal{K}_l$. For web-type, we fetch the webpage via its URL, convert it into raw text, and summarize it into a report with $\mathcal{M}_s$ as $\mathcal{K}_w$. For code-type, we meticulously search within the repository to collect file-level and function-level call graphs, analyze core code snippets, then combine them with the `README.md` file to ensemble into a report via $\mathcal{M}_s$ as $\mathcal{K}_c$.

**Iterative Query Refinement and Search.** To further uncover latent background knowledge, we task $\mathcal{M}_s$ with refining the original query based on the enriched knowledge, operating along three axes: *i)* rewriting queries that yielded insufficiently relevant results; *ii)* generalizing queries that returned overly specific results; *iii)* concretizing queries that retrieved overly broad results, formulating as:

$$\widehat{\mathcal{Q}}_{p,u} = \mathcal{M}_s(p, u, \mathcal{Q}_{p,u}, \mathcal{K}_{p,u}). \quad (7)$$

$\mathcal{K}_{p,u}$ denotes the enriched knowledge retrieved using idea part $p$ and search tool $u$. We then repeat the search of Eqn.3 with the refined query set $\widehat{\mathcal{Q}}_{p,u}$, merge the retrieved brief knowledge $\widetilde{\mathcal{K}}_j$ with the knowledge $\widetilde{\mathcal{K}}_j^*$ retained from the previous round, rank and filter the union $\widetilde{\mathcal{K}}_j \cup \widetilde{\mathcal{K}}_j^*$ via Eqn.4-6, enrich the final top-$m$ pieces, and generate a new batch of queries through Eqn.7. The entire procedure is iterated $N$ times, yielding the final enriched knowledge set $\mathcal{K}$.

### 3.2. Knowledge Grounding

For a more well-founded evaluation, we further employ a **Grounding Agent $\mathcal{M}_g$** to align ideas with knowledge at a finer granularity by excavating the specific connections between the idea $\mathcal{I}$ and the retrieved knowledge. For each $p \in \mathcal{I}$, we collect all knowledge $\mathcal{K}_p$ retrieved based on it. Then, for every $k_p \in \mathcal{K}_p$, we distill the most relevant evidences $e_p$ that genuinely supports or contradict $p$ from $k_p$ and provide a detailed relevance analysis $s_p$:

$$e_p, s_p = \mathcal{M}_g(p, k_p), \quad k_p \in \mathcal{K}_p. \quad (8)$$

Finally, we present the grounding results $\mathcal{G}$ fed into the subsequent evaluation module as:

$$\mathcal{G} = \{(p, \mathcal{G}_p)\}_{p\in\mathcal{I}}, \quad \mathcal{G}_p = \{(e_p, s_p)^i\}_{i=1}^{|\mathcal{K}_p|} \quad (9)$$

### 3.3. Multi-dimensional Multi-perspective Evaluation

**Innovation Review Board.** To achieve review consensus, we augment our evaluation module with a carefully curated innovation review board $\mathcal{P}$ specially designed for academic review. Each $\rho \in \mathcal{P}$ is an academic persona that comprises an academic profile, a knowledge familiarity vector indicating the degree of acquaintance with literature, web, and code knowledge, and reviewing habits. During evaluation, we randomly mask a proportion of knowledge corresponding to the persona's familiarity levels across different sources, thereby reflecting actual command of background knowledge. Other details about the construction, utilization, and examples of the review board are provided in Appx.F.

**Multi-dimensional Decoupled Evaluation.** We provide five initial evaluation dimensions: *Clarity*, *Validity*, *Novelty*, *Feasibility*, and *Significance*. Users can also easily register any new evaluation dimensions according to their own needs. Given the evaluation criteria set $\Psi$, for each $\psi \in \Psi$, we employ a dedicated agent evaluator $\mathcal{M}_\psi$ to assess the idea. To more faithfully emulate human peer review and improve evaluation efficiency, instead of employing all personas in $\mathcal{P}$, we randomly assign five distinct personas as reviewers to each idea. Given the selected subset $\mathcal{P}'$, for each $\rho \in \mathcal{P}'$ and $\psi \in \Psi$, we score the idea in $[0, 10]$ with a detailed review narrative that explains the underlying reasoning:

$$\varphi_{\rho,\psi} = \mathcal{M}_\psi(\rho, \mathcal{I}, \mathcal{G}). \quad (10)$$

We devise detailed evaluation guidelines and scoring rubrics for each reviewing agent to ensure rigorous.

### 3.4. Report Generation

**Point-wise Report Synthesis.** After evaluation, we synthesize all historical information to produce a comprehensive review report for the idea as formalized in Eqn.1. We directly adopt the enriched knowledge reports $\mathcal{K}$ as the background knowledge of the idea. For revision suggestions $\mathcal{V}$, the **Report Agent $\mathcal{M}_r$** proposes future improvements by combining the retrieved future information about the idea with its internal knowledge: $\mathcal{V} = \mathcal{M}_r(\mathcal{I}, \mathcal{G}_{\text{future}})$. Further, we integrate all evaluation information $\{\varphi_{\rho,\psi}\}_{\rho\in\mathcal{P}',\psi\in\Psi}$ and instruct the report agent to produce a meta-review $\varphi_{\text{meta}}$ that contains a detailed summary of the evaluation, a final score $s_{\text{point}}$, and a final decision $d_{\text{point}}$. Thus, the evaluation conclusion $E_{\text{point}}$ for this idea is expressed as follows:

$$E_{\text{point}} = \{\{\varphi_{\rho,\psi}\}_{\rho\in\mathcal{P}',\psi\in\Psi}, \varphi_{\text{meta}}\}. \quad (11)$$

The final point-wise idea report is $P_{\text{point}} = \{\mathcal{K}, \mathcal{V}, E_{\text{point}}\}$.

**Group-wise Report Synthesis.** Given a group of $n(n \geq 2)$ ideas $\{\mathcal{I}_i\}_{i=1}^n$, we first synthesize point-wise reports $P_{\text{point}}^{\mathcal{I}_i}$ for each idea. We then feed the reports of all ideas $\{P_{\text{point}}^{\mathcal{I}_i}\}_{i=1}^n$ directly as input and enable the report agent to

*Table 1.* **Quantitative Main Results.** Please refer to §4.1 and Appx.B for detailed task formulations and metric computations.

| Method | Point-wise | | | | Pair-wise | | Group-wise | | |
|---|---|---|---|---|---|---|---|---|---|
| | $Acc_2$ | $F1_2$ | $Acc_3$ | $F1_3$ | $Acc_{easy}$ | $Acc_{hard}$ | Best | Lis | Acc |
| CoT | 40.55 | 35.98 | 34.56 | 27.86 | 46.51 | 40.50 | 36.63 | 57.59 | 6.98 |
| RAG | 42.40 | 38.29 | 35.48 | 28.45 | 44.19 | 39.00 | 34.30 | 57.88 | 7.56 |
| GraphEval | 54.83 | 45.71 | 53.46 | 33.03 | 43.60 | 44.50 | 20.93 | 57.70 | 2.33 |
| ResearchAgent | 59.48 | 56.44 | 54.84 | 39.81 | 52.32 | 43.00 | 40.12 | 66.52 | 8.14 |
| InternAgent | 60.37 | 60.12 | 56.68 | 43.05 | 62.65 | 59.50 | 41.28 | 68.46 | 10.47 |
| ScholarEval | 65.44 | 65.02 | 61.75 | 58.38 | 74.42 | 60.00 | 49.42 | 70.13 | 14.53 |
| **InnoEval** | **75.58** | **75.74** | **73.73** | **74.56** | **80.81** | **63.00** | **65.12** | **76.03** | **22.09** |

compare every idea along the five aspects in $\Psi$, providing a detailed reasoning process and ultimately producing a comprehensive ranking of all ideas:

$$E_{\text{group}} = \{\{\varphi_\psi^{\text{group}}\}_{\psi \in \Psi}, \varphi_{\text{meta}}^{\text{group}}\}, \tag{12}$$

where $\varphi_\psi^{\text{group}}$ denotes the comparative analysis of different ideas under the specific dimension $\psi$, and $\varphi_{\text{meta}}^{\text{group}}$ contains the final comparative analysis together with the ranked list of ideas. The final report is $P_{\text{group}} = \{\{P_{\text{point}}^{\mathcal{I}_i}\}_{i=1}^n, E_{\text{group}}\}$.

## 4. Experimental Settings

### 4.1. Datasets and Metrics.

**Point-wise Dataset.** To curate high-quality ideas for evaluation, we extract ideas from authoritative peer-reviewed papers to construct our test dataset. Specifically, we crawl NeurIPS25 and ICLR25 papers from OpenReview and partition them into four strata according to their final decisions (Reject, Poster, Spotlight, Oral), from which we perform stratified sampling. We specially include every submission track within each stratum to ensure diversity. We apply our extraction agent $\mathcal{M}_e$ to harvest ideas from each paper, followed by manual correction and verification, finally yielding 217 point-wise samples, which we mark as $\mathcal{D}_{\text{point}}$. We design two classification tasks of different difficulty: *i) Binary classification*, predicting whether an idea can be accepted (Reject vs. others); *ii) Ternary classification*, where we group Spotlight and Oral into a single class called Highlight. We use Accuracy and macro F1 as the evaluation metrics.

**Pair-wise and Group-wise Datasets.** To construct idea groups with similar topics, we use each idea in $\mathcal{D}_{\text{point}}$ as a query to retrieve similar papers from the crawled paper pool. We select the paper with the highest similarity for each stratum to form a group, so that the ideas in each group can be automatically ranked according to their labels. We finally obtain 172 group-wise instances as $\mathcal{D}_{\text{group}}$. We define two group-wise tasks: *i) Best selection* to identify the best idea within each group, and *ii) Ranking* to produce the exact full ranking list. We evaluate best selection by Accuracy and ranking task via Longest Increasing Subsequence Match and Accuracy. Pair-wise idea comparison is also highly practical in everyday research workflows. So we sample idea pairs from $\mathcal{D}_{\text{group}}$ and split them into two difficulty levels: *i) Easy*, where the two ideas exhibit a clear label gap (e.g., Reject

vs. Highlight); *ii) Hard*, where the labels are adjacent (e.g., Poster vs. Highlight, Reject vs. Poster). After filtering, we construct a dataset $\mathcal{D}_{\text{pair}}$ comprising 372 samples, including 172 easy pairs and 200 hard pairs. We directly use Accuracy to evaluate pair-wise tasks. Further details about dataset construction and metrics are provided in Appx.B.

### 4.2. Baselines and Implementation Details

We select four categories of baselines for a comprehensive comparison with **InnoEval**. *i) Naive method*: **CoT** and **RAG**; *ii) Idea generation method*: **ResearchAgent** (Baek et al., 2025); *iii) End-to-end scientific discovery method*: **InternAgent** (Zhang et al., 2025); And specifically designed *iv) Idea evaluation method*: **GraphEval** (Feng et al., 2025) and **ScholarEval** (Moussa et al., 2025). For all baselines, we uniformly adopt DeepSeek-V3.2 (DeepSeek-AI, 2025) as the backbone model. We also test with o4-mini (OpenAI, 2025) in §5.2 to demonstrate **InnoEval**'s robustness across different models. Please refer to Appx.C for more details about the baselines and our reproduction.

**Implementation.** We use `bge-base-en-v1.5` as the retriever and `bge-reranker-base` as the reranker (Xiao et al., 2024) for the retrieval in our paper. At search time, we set the maximum retention number $m$ for each resource type to 10, fix the weight $\alpha$ between semantic score and LLM score at 0.2, and cap the rounds of query refinement $N$ to 3. The cost of evaluating one sample is 0.42$. All the prompts used in our paper can be found in Appx.H.

## 5. Experiment Results

### 5.1. Main Results

**Quantitative Results.** As shown in Tab.1, our **InnoEval** can achieve state-of-the-art performance compared to other baselines across all tasks. On the point-wise tasks, we observe label collapse on most baselines: their predictions only concentrate on one or two labels, manifested by F1 scores substantially lower than Accuracy. Owing to ample evidence support and multi-dimensional, multi-perspective evaluation, our method disperses label predictions and attains F1 scores of 75.74% and 74.56%. This also enables **InnoEval** to perform better on pair and group tasks. Since GraphEval is trained only for single-idea label prediction, its performance on pair and group tasks is very poor, highlighting the flexibility requirements of the idea evaluation system. Additionally, ResearchAgent relies on a pre-constructed literature collection, making its evaluation of relatively new ideas inadequate. Although InternAgent and ScholarEval design sophisticated search pipelines, due to limitations in search resources and the singularity of evaluation, their evaluation results struggle to align with human labels.

**Qualitative Results.** Beyond quantitative tests, we also

*Table 2.* **Qualitative Main Results** by using o4-mini as the judge. *Rationality* measures the logical soundness; *Supportiveness* indicates whether the analysis is adequately backed by evidence; *Depth* assesses the thoroughness of the review; *Constructiveness* reflects the usefulness and feasibility of the suggestions; *Overall Quality* provides a holistic judgment of the entire content.

| Baselines | Rationality | | Supportiveness | | Depth | | Constructiveness | | Overall Quality | |
|---|---|---|---|---|---|---|---|---|---|---|
| **InnoEval vs.** | Win(%) | Lose(%) | Win(%) | Lose(%) | Win(%) | Lose(%) | Win(%) | Lose(%) | Win(%) | Lose(%) |
| CoT | **88.48** | 9.22 | **92.17** | 2.76 | **93.09** | 6.91 | **89.77** | 9.77 | **90.70** | 7.83 |
| RAG | **87.10** | 11.98 | **87.56** | 11.52 | **92.63** | 6.45 | **87.10** | 9.68 | **90.32** | 8.37 |
| ResearchAgent | **86.18** | 12.44 | **86.18** | 7.37 | **90.32** | 8.29 | **88.94** | 9.68 | **89.86** | 8.29 |
| InternAgent | **83.41** | 15.67 | **87.10** | 9.22 | **91.24** | 7.37 | **82.03** | 16.59 | **85.71** | 12.90 |
| ScholarEval | **67.28** | 28.11 | **61.75** | 32.72 | **70.51** | 28.11 | **84.79** | 14.29 | **71.89** | 25.81 |

*Figure 2.* **Left Heat Map: Human Evaluation.** Correlations between `InnoEval`'s scores on the five dimensions and the scores assigned by human experts (*Human*), as well as by online peer-review comments (*Reviews*) on the same five dimensions. **Right Bar Charts: Ablation Studies.** *-Grounding* removes the grounding module and feeds raw search results directly into evaluation; *-Personalized* disables the persona, letting the agent evaluate without assumed identity; *-Web&Code* restricts retrieval to relevant papers only (no web or code search); *o4-mini* swaps the backbone model for o4-mini; *InnoEval* denotes our full configuration.

qualitatively compare the point-wise evaluation quality of `InnoEval` and other baselines from multiple perspectives using LLM-as-judge, as reported in Tab.2. Our multi-source, in-depth search renders the evaluation results more reasonable (*Rationality*) and well-grounded (*Supportiveness*). Meanwhile, the multi-dimensional and multi-perspective evaluation paradigm grants our analysis an over 90% win-rate in *Depth* compared with most methods. In addition, the improvement suggestions derived from future papers endow our evaluation with stronger *Constructiveness*, with over 80% win-rate compared with all baselines. It is worth noting that ScholarEval is a strong baseline, which beats `InnoEval` on more than 25% of the samples in terms of *rationality*, *supportiveness*, and *depth*. Nevertheless, the limited evaluation dimensions and the lack of evidence-based improvement suggestions render its *constructiveness* comparable to that of other baselines. Overall, the comprehensive and systematic evaluation report produced by `InnoEval` yields an over 70% win-rate in *Overall Quality*.

**Human Evaluation.** We randomly sample 60 instances from $\mathcal{D}_{point}$ and have human experts score them on five dimensions under the same rating scheme with `InnoEval`. We also prompt an LLM to score them on the five dimensions, conditioned on their ground-truth peer-review comments, to represent real peer reviews. Detailed evaluation procedures are provided in Appx.D, with all results shown in Fig.2. It can be observed that `InnoEval`'s scores exhibit strong positive correlations with both expert and peer-review judgments across all five dimensions, with correlation coefficients $\geq 0.5$. Among the five dimensions, clarity yields the

highest correlation, as it solely concerns logical and structural coherence and is therefore relatively straightforward to assess. In contrast, significance shows relatively lower correlation, reflecting its inherent complexity and thus points to a promising avenue for our future work.

## 5.2. Ablations

In Fig.2, we illustrate `InnoEval`'s performance after ablating or replacing key modules. It can be seen that removing grounding (*-Grounding*) leads to varying degrees of performance degradation across different tasks, indicating that our Grounding Agent can effectively filter out noise in the retrieved reports and enables the evaluation to focus more on relevant information. Moreover, directly employing LLM-as-Judge (*-Personalized*) causes a marked performance drop, especially on point-wise and group-wise tasks, underscoring that incorporating personalized evaluation effectively mitigates the bias and subjectivity inherent in LLM-as-Judge. To validate the effectiveness of leveraging diverse retrieval sources, we re-conduct the evaluation using only the literature search results (*-Web&Code*). The outcome shows that search richness can significantly improve evaluation accuracy and emphasizes the critical role of sufficient background knowledge in idea assessment. A finer-grained observation is that restricting retrieval to literature gives more hurt to pair-wise and group-wise tasks, indicating that comparing multiple ideas demands especially rich background knowledge. We further replace `InnoEval`'s backbone from DeepSeek-V3.2 with o4-mini. Despite slight declines on some tasks, we can still surpass the strongest baseline,

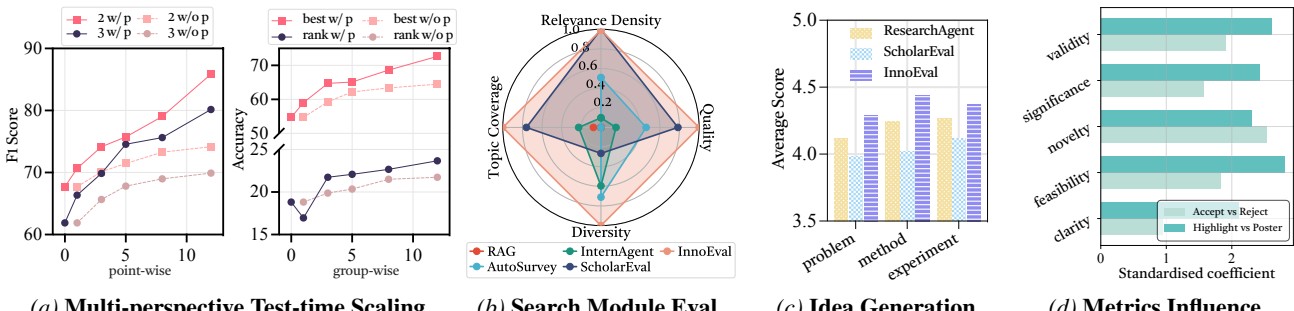

*(a)* **Multi-perspective Test-time Scaling.** *(b)* **Search Module Eval.** *(c)* **Idea Generation.** *(d)* **Metrics Influence.**

*Figure 3. (a)* **Multi-perspective Test-time Scaling.** We compare the test-time scaling results with or without academic personas on point-wise and group-wise tasks. *(b)* **Search Module Eval.** We compare our heterogeneous deep knowledge search engine with the search modules of other baselines from four metrics. The specific definition of each metric can be found in Appx.E. *(c)* **Idea Generation.** We use the evaluation results of different methods as feedback to improve the idea generation pipeline in ResearchAgent. *(d)* **Metrics Influence.** We explore critical metrics determining acceptance or highlighting of the idea by linear regression.

demonstrating its robustness across different models.

### 5.3. Analysis

**Multi-perspective Test-time Scaling.** In Fig.3a, we show how `InnoEval`'s performance varies with the number of different academic personas in evaluation, compared with vanilla Test-Time Scaling (TTS) without personas. Clear scaling laws are observed under both *w/* and *w/o* persona, demonstrating that increasing generation diversity can effectively enhance evaluation performance. However, across all task settings, plain TTS consistently underperforms personalized TTS, and as the number of samples increases, the performance gain of vanilla TTS gradually plateaus, whereas personalized TTS maintains robust growth momentum. This further underscores the importance of reviewer consensus. Vanilla TTS essentially coerces a single reviewer to fabricate divergent opinions, whereas authentic consensus emerges from reviewers with diverse backgrounds converging after examining the idea from distinct perspectives. Another interesting phenomenon can corroborate this conclusion. In group-wise setting, using a single persona underperforms the no-persona baseline on ranking tasks, as the shift from 0 to 1 persona merely transforms the inherent bias of the LLM to that specific persona, without addressing the fundamental issue. Through this empirical study, it should be noted that our choice of randomly selecting 5 personas in the main experiments balances evaluation efficiency and effectiveness, as a larger pool would sharply increase inference time, but it does not represent the performance ceiling of `InnoEval`.

**Effectiveness of Deep Knowledge Search Engine.** To further demonstrate the effectiveness of our deep knowledge search engine, in Fig.3b we specially isolate and compare the search engines employed by each baseline, additionally including AutoSurvey (Wang et al., 2024b), a baseline specifically designed for literature surveys with a strong retrieval engine. Although our two strong baselines ScholarEval achieve high relevance, it causes the search results to converge and thereby sacrifice diversity. AutoSurvey, in pur-

suit of diverse results, neglects coverage of the idea's topic. Only our heterogeneous deep knowledge search engine simultaneously preserves relevance, coverage, and diversity.

**`InnoEval`'s reviews can serve as an effective reference for idea optimization.** In Fig.3c, we integrate different evaluation methods into the idea iteration pipeline of ResearchAgent (an idea generation method) and rigorously adhere to the ResearchAgent's benchmark and setting to assess the quality of the generated ideas. The results demonstrate that the actionable revision suggestions provided by `InnoEval` further boost idea quality throughout the iterative process, yielding significant improvements across problem formulation, methodology, and experimental design compared to the vanilla ResearchAgent. This improvement is attributed to our multi-dimensional idea evaluation and highly feasible revision feedback in the evaluation reports. In contrast, ScholarEval's exclusive focus on the contribution and soundness dimensions leads to biased idea optimization, resulting in degraded generation quality.

**Novelty is the most critical factor in an idea's acceptance, whereas achieving highlight requires well-rounded development across all dimensions.** In Fig.3d, we try to explore what is the most critical dimension that determines whether an idea will be accepted or highlighted. We frame it as a linear regression problem over five evaluation dimensions. The acceptance of an idea can thereby be cast as a binary classification of *reject* versus other labels (*poster* or *highlight*), while the elevation to highlight can be further reduced to a second binary distinction between *poster* and *highlight*. We fit a linear regression on $\mathcal{D}_{point}$ predicted by `InnoEval` and interpret the resulting coefficient magnitudes as indicators of dimensional importance. We can see that *novelty* is the most decisive predictor of whether an idea is accepted, a result consistent with human intuition. *Validity*, *significance*, and *feasibility* are also influential, whereas *clarity* exerts the weakest effect. However, once the acceptance bar is reached, *feasibility* gains prominence. This means the focus shifts to whether comprehensive experiments can

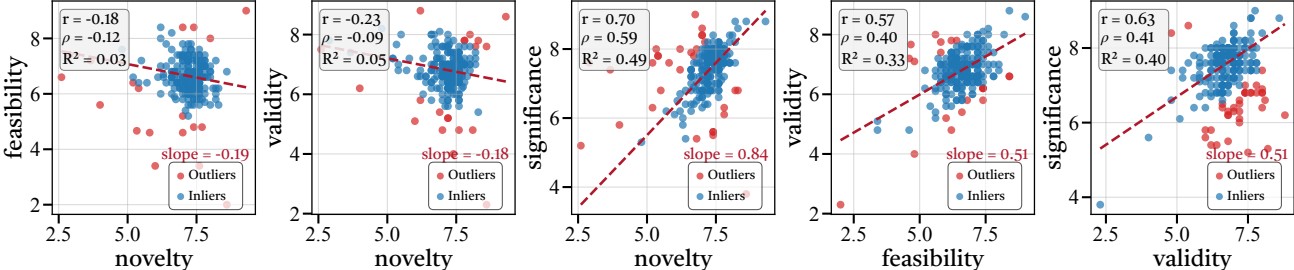

*Figure 4.* **Scatter Plots Between Metric Pairs.** We perform linear regression fitting for each metric pair. The red dashed line is the fit after removing outliers, and its *slope* is reported. $r$ denotes the Pearson coefficient, $\rho$ is the Spearman coefficient, and $R^2$ represents the fit goodness of all inliers explained by the fitted line. A complete version of all metric pairs' correlation can be seen in Fig.5.

be designed to demonstrate the proposed methodology. All remaining dimensions likewise carry substantial weight, indicating that highlight status requires well-rounded strength and underscores the demand for a solid contribution.

**Mutual relationships among evaluation dimensions.** To further examine the interrelations among all evaluation dimensions, we plot pairwise scatter plots of all the metrics and fit their linear trends, then quantify their correlations as shown in Fig.4. Several interesting findings deserve emphasis: 1) *Significance* exhibits strong positive correlations with both *novelty* and *validity*, indicating that creative ideas grounded in solid theory can exert lasting impacts. 2) *Feasibility* and *validity* are likewise strongly correlated, an expected pattern aligning with human cognition: theoretically well-founded ideas are easier to verify experimentally. 3) *Novelty* shows slight negative correlations with *validity* and *feasibility*, suggesting that more novel ideas are less likely to receive theoretical support or experimental confirmation. 4) Only *clarity* and *significance* exhibit positive correlations with all other metrics, which also makes sense because *clarity* serves as the prerequisite for every other dimension, whereas *significance* relies on the remaining metrics as its foundation. These observations align well with scholarly cognition, providing convergent evidence that InnoEval can successfully capture the essence of idea evaluation.

**InnoEval can recognize the real innovation in practice.** In Appx.G, we show InnoEval's evaluation report of a famous idea from "*Mamba: Linear-Time Sequence Modeling with Selective State Spaces*" (Gu & Dao, 2023). Our search engine can retrieve key references for Mamba (e.g., S4 (Gu et al., 2022), FlashAttention-V2 (Dao, 2024), H3 (Fu et al., 2023)), relevant discussion blogs from the web, and important code repos related to its experiments (e.g., state-space/s4[1]). After fine-grained grounding, reviewers with diverse academic backgrounds and preferences evaluate the idea from multiple perspectives and dimensions, with each dimension containing detailed review comments. The final meta-review synthesizes all opinions to provide an overall assessment and decision. Notably, consensus across different perspectives can effectively mitigate biases inher-

ent to single viewpoints (e.g., Reviewer 2), yielding sound decisions and preventing the tragedy of genuinely innovative ideas being overlooked in practice. Finally, the report includes actionable suggestions for idea improvement from multiple angles, including method, model, experiments, etc.

## 6. Related Work

**LLMs for Scientific Discovery.** The rapidly advancing reasoning abilities of LLMs (Qiao et al., 2023; Chen et al., 2025b) have recently enabled their increasingly prominent role in Automated Scientific Discovery (Nie et al., 2026; Schmidgall et al., 2025; Lu et al., 2024; Xie et al., 2025; Qiao et al., 2025), which contains *Literature Reviewing* (Wang et al., 2024b; Liang et al., 2025; Asai et al., 2024; Yan et al., 2025; Agarwal et al., 2025), *Idea Generation* (Li et al., 2024; Su et al., 2025; Zhao et al., 2025; Baek et al., 2025; Wang et al., 2024a), *Method Implementation* (Novikov et al., 2025; Ou et al., 2025; Liu et al., 2025; Yu et al., 2025; Jiang et al., 2025), and *Manuscript Writing* (Wen et al., 2024, Chen et al., 2025a). Nevertheless, while numerous efforts are pursuing the whole automated discovery pipeline (Lu et al., 2024; Schmidgall et al., 2025; Weng et al., 2025; Yamada et al., 2025; Shao et al., 2025; Zhang et al., 2025), very few works have devoted full attention to the evaluation of generated ideas. Because innovation constitutes the core of scientific discovery, its quality deterministically conditions every downstream phase, rendering systematic assessment of ideas both urgent and paramount.

**LLMs for Idea Generation and Evaluation.** Knowledge is an indispensable part for idea generation and evaluation, because rich idea-related background knowledge is the guarantee for their effective operation. Prior efforts rely mainly on LLMs' internal parametric knowledge (Lu et al., 2024; Jiang et al., 2025; Novikov et al., 2025; Yu et al., 2025; Yamada et al., 2025), which is inherently insufficient. Although recent work permits LLMs to invoke paper-search tools (Baek et al., 2025; Su et al., 2025; Wang et al., 2025; 2024a; Si et al., 2025; Li et al., 2024; Zhao et al., 2025; Ju et al., 2025; Tang et al., 2025; Zhang et al., 2025; 2026), these methods remain constrained by the limited breadth and depth of the searched resources, leading to narrow knowl-

---

[1] https://github.com/state-spaces/s4.

edge horizons. Besides, current evaluation rely on simple LLM-as-a-Judge (Tang et al., 2025; Zhang et al., 2025; Si et al., 2025; Shahid et al., 2025; Moussa et al., 2025), introducing systematic bias and subjectivity that overlooks the review consensus needed by a fair scientific evaluation.

## 7. Conclusion

We introduce `InnoEval`, a deep idea evaluation framework to achieve multi-dimensional, multi-perspective innovation assessment grounded in heterogeneous knowledge. We construct an idea evaluation dataset that supports pointwise, pair-wise, and group-wise assessment, incorporating quantitative, qualitative, and human evaluation strategies. We design a comprehensive suite of experiments to analyze `InnoEval`'s performance and behavior, providing insightful findings for the community's future research.

## Impact Statement

Despite its potential to accelerate innovation evaluation, reduce human effort, and optimize academic resource allocation, the development of `InnoEval` necessitates careful attention to ethical and societal considerations. Although we strive to mitigate LLM-as-a-Judge biases and align with human evaluation preferences, LLM-generated content may inevitably contain hallucinations and unreliable information. Therefore, we do not advocate over-reliance on it. Instead, we promote a human-AI collaborative paradigm where it serves as an auxiliary tool for human decision-making, augmenting rather than replacing human expertise. Users should carefully scrutinize its generated content. We will open-source our code framework and evaluation data to enhance system transparency.

## Acknowledgments

We would like to express our sincere gratitude to the anonymous reviewers for their thoughtful and constructive feedback. This work was supported by the National Natural Science Foundation of China (No. 62576307), the Fundamental Research Funds for the Central Universities (226-2023-00138), the Yongjiang Talent Introduction Programme (2021A-156-G), the CIE-Tencent Doctoral Research Incentive Program (Hunyuan Large Model Special Program), and the Information Technology Center and State Key Lab of CAD&CG, Zhejiang University.

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

## A. Limitations

This paper still has some limitations that must be acknowledged: *a)* **Limited Discipline**. Currently, we focus exclusively on innovation evaluation within the AI domain. In the future, we plan to extend our scope to disciplines including biology, medicine, physics, geography, oceanography, etc. *b)* **Efficiency**. Due to our multi-source heterogeneous deep search and multi-perspective multi-dimensional evaluation methodology, evaluating a single sample requires about half an hour. However, our method supports large-scale parallel evaluation, making it highly efficient for batch processing (approximately 100 samples per hour). *c)* **Limited Modality**. Now we are limited to text-based ideas. However, ideas can also be expressed in other modalities such as flowcharts, slides, and videos. In the future, we will explore evaluation for additional modalities.

## B. Dataset

### B.1. Point-wise Dataset

**Dataset Construction.** We collect papers via the official OpenReview API[2]. Specifically, we crawl NeurIPS 2025 and ICLR 2025 papers from OpenReview, filtering out withdrawn submissions without review comments and placeholder papers lacking full text. The remaining papers are partitioned into four strata according to their final decisions (Reject, Poster, Spotlight, Oral), from which we perform stratified sampling. We specially include every submission track in the conference within each stratum to ensure idea diversity. We apply our extraction agent $\mathcal{M}_e$ to harvest ideas from each paper, followed by manual correction and verification, finally yielding 217 point-wise samples, which we mark as $\mathcal{D}_{\text{point}}$. The final dataset comprises 136 ICLR 2025 ideas and 81 NeurIPS 2025 ideas, including 138 Reject, 66 Poster, 9 Spotlight, and 4 Oral decisions. Consequently, Reject, Poster, and Highlight (Spotlight + Oral) ideas account for 61.3%, 29.3%, and 9.4% of all the samples, respectively. Their detailed track distributions are listed in Tab.4.

**Tasks and Metrics.** We have two classification tasks of different difficulty in this setting: *i) Binary classification*, predicting whether an idea can be accepted. The labels are Reject and Accept (including Poster, Spotlight, and Oral); *ii) Three-way classification*, where we group Spotlight and Oral into a single class called Highlight. The labels are Reject, Poster, and Highlight (including Spotlight and Oral). We use Accuracy and macro F1 score as the evaluation metrics.

### B.2. Group-wise Dataset

**Dataset Construction.** To construct idea groups with similar topics, we use the abstract of each idea in $\mathcal{D}_{\text{point}}$ as a query to retrieve similar papers from all the papers we crawl from ICLR 2025 and NeurIPS 2025. We adopt `bge-base-en-v1.5`[3] as the embedding model to retrieve the 800 most similar papers, then apply `bge-reranker-base`[4] as the reranker to reduce the pool to 120. From these 120 candidates, we select the highest-similarity paper in each label stratum to form a group, enabling automatic ranking of the ideas by their labels. We apply the same extraction and verification pipeline to the retrieved papers, finally obtaining 172 group-wise instances as $\mathcal{D}_{\text{group}}$.

**Tasks and Metrics.** We define two objectives for group-wise tasks: *i) Best Selection* to identify the single best idea within each group, and *ii) Ranking* to produce the exact full ranking list of all ideas in the group. We evaluate the best-idea selection by Accuracy, and assess the ranking task via Longest Increasing Subsequence (LIS) score as well as Accuracy. The LIS score is computed as follows: let the group size be $n$ and the length of the longest increasing subsequence of the predicted ranking with respect to the gold ranking be $l$. Then the LIS score is $\frac{l}{n}$. For example, if the gold ranking is $\{1, 2, 3, 4\}$ and the predicted ranking is $\{2, 1, 3, 4\}$, their longest increasing subsequence is $\{1, 3, 4\}$ or $\{2, 3, 4\}$, all with a length of 3, so the LIS score is $\frac{3}{4} = 0.75$. And for *ranking* tasks, the Accuracy$= 1$ if and only if the predicted ranking is identical to the gold ranking, otherwise the Accuracy will be 0.

### B.3. Pair-wise Dataset

Since idea pairs naturally exist within each group, our pairwise dataset is constructed on top of $\mathcal{D}_{\text{group}}$. For a group of size $n$, we can sample $\binom{n}{2}$ pairs. We perform the sampling at two difficulty levels: 1) *easy*: pairs whose two ideas have markedly different labels (Highlight vs. Reject), and are therefore easier to distinguish. 2) *difficult*: pairs whose two ideas have similar

---

[2] https://api2.openreview.net.
[3] https://huggingface.co/BAAI/bge-large-en-v1.5
[4] https://huggingface.co/BAAI/bge-reranker-base

*Table 3.* Notations used in our paper.

| Notation | Description |
|---|---|
| **Problem Definition (§2)** | |
| $\mathcal{I}$ | A structured idea contains six parts. |
| $t$ | The evaluation timestamp for an idea. |
| $\mathcal{F}$ | Evaluation system. |
| $P$ | A final evaluation report. |
| $P_{\text{point}}$ | Point-wise evaluation report. |
| $\mathcal{K}$ | Background knowledge of the idea. |
| $E_{\text{point}}$ | Point-wise evaluation results. |
| $\mathcal{V}$ | Revision suggestions for future improvements of the idea. |
| $d_{\text{point}}$ | Final decision of an idea. |
| $\mathcal{I}_i$ | The $i$-th idea of a group of $n$ ideas. |
| $P_{\text{point}}^{\mathcal{I}_i}$ | Evaluation report for idea $\mathcal{I}_i$. |
| $P_{\text{group}}$ | Group-wise evaluation report with $\geq 2$ ideas. |
| $E_{\text{group}}$ | Group-wise evaluation results. |
| $d_{\text{group}}$ | Ranking list of a group of ideas. |
| **Heterogeneous Deep Knowledge Search (§3.1)** | |
| $T$ | Raw textual idea. |
| $\mathcal{M}_e$ | Extraction agent used to extract $T$ into structured parts $\mathcal{I}$. |
| $\mathcal{M}_s$ | Search agent as the backbone of our search engine. |
| $p$ | Idea part $p \in \mathcal{I}$. |
| $u$ | Search tool $u \in (\mathsf{X}, \mathsf{\nabla}, \mathsf{8}, \mathsf{G}, \mathsf{\Omega}, \mathsf{k})$. |
| $\mathcal{Q}_{p,u}$ | Search queries specially for idea part $p$ and search tool $u$. |
| $\widetilde{\mathcal{K}}$ | Brief searched knowledge. |
| $\widetilde{\mathcal{K}}_{p,u}$ | Brief knowledge searched by search tool $u$ based on $p$. |
| $\widetilde{\mathcal{K}}_l, \widetilde{\mathcal{K}}_w, \widetilde{\mathcal{K}}_c$ | Brief searched literature, web, code knowledge respectively. |
| $m$ | Maximum retention number for each knowledge type. |
| $\mathcal{S}_j^{\text{sem}}$ | Ranking scores for knowledge type $j \in \{l, w, c\}$ calculated through semantic similarity. |
| $\mathcal{S}_j^{\text{llm}}$ | Ranking scores for knowledge type $j \in \{l, w, c\}$ given by LLM. |
| $\mathcal{S}_j$ | Overall ranking scores for knowledge type $j \in \{l, w, c\}$ weighted by $\mathcal{S}_j^{\text{sem}}$ and $\mathcal{S}_j^{\text{llm}}$ |
| $\widetilde{\mathcal{K}}_j^*$ | Top-$m$ brief knowledge for knowledge type $j \in \{l, w, c\}$ ranked by $\mathcal{S}_j$ |
| $\mathcal{K}_l, \mathcal{K}_w, \mathcal{K}_c$ | Enriched searched literature, web, code knowledge reports respectively. |
| $\mathcal{K}_{p,u}$ | Enriched knowledge retrieved using idea part $p$ and search tool $u$. |
| $\widehat{\mathcal{Q}}_{p,u}$ | Refined queries specially for idea part $p$ and search tool $u$. |
| $N$ | Query refinement iteration times. |
| **Knowledge Grounding (§3.2)** | |
| $\mathcal{M}_g$ | Grounding agent used to ground searched knowledge to idea part at a finer-grained degree. |
| $\mathcal{K}_p$ | Knowledge reports searched based on idea part $p$. |
| $k_p$ | Knowledge piece $k_p \in \mathcal{K}_p$. |
| $e_p$ | Knowledge evidences extracted based on $k_p$ and $p$. |
| $s_p$ | Relevance analysis between $e_p$ and $p$ |
| $\mathcal{G}_p$ | Grounding results for idea part $p$. |
| $\mathcal{G}$ | Grounding results for all idea parts. |
| $\mathcal{G}_{\text{future}}$ | Grounding results from future knowledge. |
| **Multi-dimensional Multi-perspective Evaluation (§3.3)** | |
| $\mathcal{P}$ | Innovation review board. |
| $\rho$ | Academic persona $\rho \in \mathcal{P}$. |
| $\Psi$ | Evaluation dimension set. |
| $\psi$ | Evaluation dimension $\psi \in \Psi$. |
| $\mathcal{M}_\psi$ | Evaluator agent specially for evaluating $\psi$. |
| $\mathcal{P}'$ | Chosen personas during evaluation. |
| $\varphi_{\rho,\psi}$ | Evaluation result on dimension $\psi$ by reviewer $\rho$. |
| **Report Generation (§3.4)** | |
| $\mathcal{M}_r$ | Report agent for report generation. |
| $\varphi_{\text{meta}}$ | Meta-review. |
| $\varphi_\psi^{\text{group}}$ | Comparative evaluation of group ideas under dimension $\psi$. |
| $\varphi_{\text{meta}}^{\text{group}}$ | Meta-review for group evaluation. |

*Table 4.* Detailed track distribution of our point-wise dataset.

| ICLR 2025 | | NeurIPS 2025 | |
|---|---|---|---|
| 1. foundation or frontier models, including LLMs | 15 | 1. Deep learning (e.g., architectures, generative models, optimization for deep networks, foundation models, LLMs) | 24 |
| 2. generative models | 13 | 2. Theory (e.g., control theory, learning theory, algorithmic game theory) | 11 |
| 3. applications to computer vision, audio, language, and other modalities | 11 | 3. Machine learning for sciences (e.g. climate, health, life sciences, physics, social sciences) | 7 |
| 4. reinforcement learning | 11 | 4. Applications (e.g., vision, language, speech and audio, Creative AI) | 7 |
| 5. datasets and benchmarks | 10 | 5. Probabilistic methods (e.g., variational inference, causal inference, Gaussian processes) | 6 |
| 6. interpretability and explainable AI | 10 | 6. Social and economic aspects of machine learning (e.g., fairness, interpretability, human-AI interaction, privacy, safety, strategic behavior) | 6 |
| 7. unsupervised, self-supervised, semi-supervised, and supervised representation learning | 10 | 7. Reinforcement learning (e.g., decision and control, planning, hierarchical RL, robotics) | 5 |
| 8. applications to physical sciences (physics, chemistry, biology, etc.) | 7 | 8. Optimization (e.g., convex and non-convex, stochastic, robust) | 4 |
| 9. learning theory | 7 | 9. Neuroscience and cognitive science (e.g., neural coding, brain-computer interfaces) | 3 |
| 10. applications to robotics, autonomy, planning | 6 | 10. General machine learning (supervised, unsupervised, online, active, etc.) | 3 |
| 11. learning on graphs and other geometries & topologies | 6 | 11. Infrastructure (e.g., libraries, improved implementation and scalability, distributed solutions) | 2 |
| 12. alignment, fairness, safety, privacy, and societal considerations | 6 | 12. Others | 2 |
| 13. learning on time series and dynamical systems | 5 | 13. Evaluation (e.g., methodology, meta studies, replicability and validity, human-in-the-loop) | 1 |
| 14. other topics in machine learning (i.e., none of the above) | 5 | | |
| 15. optimization | 4 | | |
| 16. applications to neuroscience & cognitive science | 4 | | |
| 17. neurosymbolic & hybrid AI systems (physics-informed, logic & formal reasoning, etc.) | 3 | | |
| 18. causal reasoning | 1 | | |
| 19. infrastructure, software libraries, hardware, systems, etc. | 1 | | |
| 20. transfer learning, meta learning, and lifelong learning | 1 | | |

labels (Poster vs. Reject, Highlight vs. Poster), and are therefore harder to distinguish. After screening and filtering, we construct a dataset $\mathcal{D}_{\text{pair}}$ comprising 372 samples, including 172 easy pairs and 200 difficult pairs. We directly use Accuracy to evaluate pair-wise tasks and report the results of easy and difficult pairs separately.

## C. Baselines and Reproduction Details

We select four categories of baselines for a comprehensive comparison with **InnoEval**.

*i) Naive method*:

- **CoT** that relies solely on the model's internal knowledge to assess and decide on ideas. We adopt a strong implementation for CoT. We directly use our evaluation agent $\mathcal{M}_\psi$ in InnoEval to assess each idea across all dimensions by simply discarding both the grounding results and the personalized settings. Then we use the report agent $\mathcal{M}_r$ to produce the final decision based on the evaluation results.
- **RAG** that directly retrieves relevant papers as references. Building upon CoT, we use our search agent $\mathcal{M}_s$ to directly fast retrieve (not including slow search and query refinement) top-$m$ relevant papers from arXiv and incorporate the meta-information (including abstracts) of all retrieved papers into $\mathcal{M}_\psi$ for reference during evaluation.

*ii) Idea generation method*:

- **ResearchAgent** (Baek et al., 2025), a representative idea generation method which incorporates a reviewing agent for idea refinement. We follow the reproduction details in Feng et al. (2025) where the research problem, scientific method, and experiment design of an idea are each scored by a reviewing agent based on five criteria. Building on this, they further introduce a final decision step that synthesizes the above evaluation results to provide a comprehensive review decision. In our reproduction, we further optimize the prompt for the final decision step to better align with our point-wise tasks. We also use our carefully designed prompt in $\mathcal{M}_r$ to give pairwise and group-wise results based on ResearchAgent's own evaluation results.

*iii) End-to-end scientific discovery method*:

- **InternAgent** (Zhang et al., 2025), a representative end-to-end scientific-discovery framework in which idea assessment is one integral component. We directly employ its survey module to retrieve relevant literature and its assessment module to perform the evaluation. We adapt their outputs to our tasks using the same strategy as employed for the previous baselines.

*iv) Idea evaluation method*:

- **GraphEval** (Feng et al., 2025), which employs graph propagation to predict idea labels. We follow the same settings in the GraphEval paper and use its official code to train a GNN model and conduct label prediction on our point-wise dataset. For both group-wise and pairwise tasks, we rank ideas directly according to their predicted labels. When multiple ideas are assigned the same label, we take the predicted probability from the model's final classification layer as the confidence score and use this confidence to break ties.
- **ScholarEval** (Moussa et al., 2025), which focuses on assessing the soundness and contribution of ideas grounded in literature. We run the whole ScholarEval pipeline to synthesize soundness and contribution reviews and adapt these reviews to final decisions like other baselines.

For all baselines, we uniformly adopt DeepSeek-V3.2 (DeepSeek-AI, 2025) as the backbone model. We also test with o4-mini (OpenAI, 2025) as the backbone in §5.2 to demonstrate InnoEval's robustness across different backbones.

## D. Human Evaluation

**Human Scoring.** We invite five domain experts at the forefront of AI (3 Ph.D. students, 1 professor, and 1 algorithm engineer) to score the 60 sampled instances on five dimensions: clarity, novelty, feasibility, validity, and significance, following the identical scoring scheme used by InnoEval. During scoring, the experts are permitted to employ any search tools to augment their background knowledge and assign ratings based on the searched knowledge combined with their prior domain expertise. We calculate the Pearson correlation between the dimension-wise mean scores provided by the human experts and the corresponding scores produced by InnoEval.

**Review Scoring.** To further mirror authentic human peer-review judgments, we crawl the publicly available peer-review records for the 60 sampled instances, including individual reviewers' comments and the meta-reviews. Although these reviews rarely contain explicit dimension-wise scores, the five aspects are implicitly encoded in the textual discourse. We therefore employ an LLM to extract these latent cues and assign concrete scores on each of the five dimensions, strictly adhering to the same scoring protocol used by InnoEval. Pearson correlations between the LLM-extracted scores and the InnoEval scores are subsequently computed.

## E. Search Metrics

Here, we detail the computation of each metric shown in Fig.3b. For an idea $\mathcal{I}$, let $\mathcal{R}$ denote the set of all resources retrieved by a method, and $\mathcal{R}_h \subseteq \mathcal{R}$ the subset of highly relevant resources. Let $\mathcal{T}_{\mathcal{I}}$ be the main topic words contained in idea $\mathcal{I}$, and $\mathcal{T}_{r_i}$ the main topic words contained in each resource $r_i \in \mathcal{R}, 1 \le i \le |\mathcal{R}|$. The metrics are then defined as follows:

- **Relevance Density.** This metric quantifies the relevance degree of retrieved resources. We use an LLM to determine highly relevant sources $\mathcal{R}_h$ returned by each method. To prevent methods from winning simply by retrieving more total items, we divide the count of highly relevant resources $|\mathcal{R}_h|$ by the total number retrieved $|\mathcal{R}|$, yielding the relevance density of each search strategy:

$$\text{Relevance Density} = \frac{|\mathcal{R}_h|}{|\mathcal{R}|}. \tag{13}$$

- **Topic Coverage.** This metric measures the comprehensiveness of the retrieved resources, specifically the extent to which

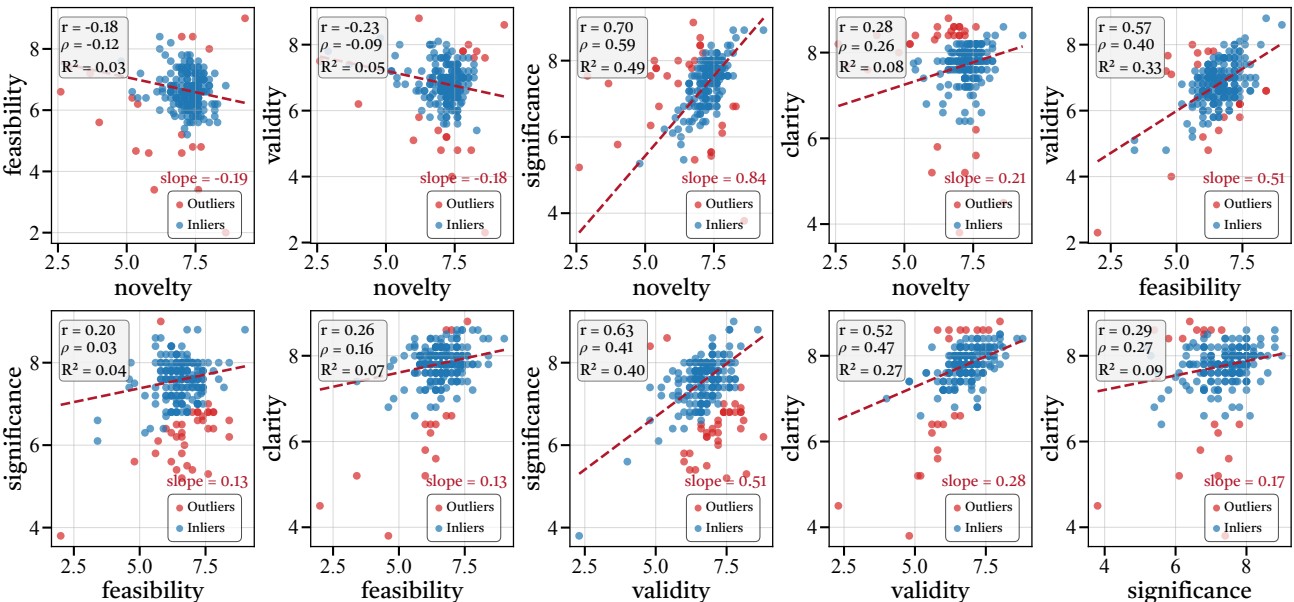

*Figure 5.* **Scatter Plots Between Metric Pairs.** We perform linear regression fitting for each metric pair. The red dashed line is the fit after removing outliers, and its *slope* is reported. $r$ denotes the Pearson coefficient, $\rho$ is the Spearman coefficient, and $R^2$ represents the fit goodness of all inliers explained by the fitted line.

the topic keywords covered by the retrieved resources overlap with those of the idea. It is defined as follows:

$$\textbf{Topic Coverage} = \frac{|\mathcal{T}_{\mathcal{I}} \cap (\mathcal{T}_{r_1} \cup \cdots \cup \mathcal{T}_{r_{|\mathcal{R}|}})|}{|\mathcal{T}_{\mathcal{I}}|} \tag{14}$$

- **Diversity.** This metric quantifies the diversity of the retrieved resources, expressed as the proportion of topic keywords beyond those already covered in the idea that appear in the retrieved documents:

$$\textbf{Diversity} = \frac{|\mathcal{T}_{r_1} \cup \cdots \cup \mathcal{T}_{r_{|\mathcal{R}|}} - \mathcal{T}_{\mathcal{I}}|}{|\mathcal{T}_{r_1} \cup \cdots \cup \mathcal{T}_{r_{|\mathcal{R}|}}|} \tag{15}$$

- **Quality.** We directly employ an LLM to assess the quality of each retrieved resource and assign it a score $s_{r_i}$. The final metric is the average quality across all resources:

$$\textbf{Quality} = \frac{\sum_{i=1}^{|\mathcal{R}|} s_{r_i}}{|\mathcal{R}|} \tag{16}$$

All LLMs employed in the above computations are DeepSeek-V3.2.

## F. Innovation Review Board

To mitigate LLM-as-Judge bias and align evaluations more closely with human preferences, we curate a domain-specific persona base tailored to academic reviewing. Each persona represents a routine AI practitioner, such as professors, PhD students, algorithm engineers, etc. Specifically, each persona $\rho$ in our persona base $\mathcal{P}$ contains the following information:

- **Background** outlines the persona's academic background, research preferences, and reviewing tendencies.
- **Background Knowledge** quantifies the persona's command of relevant background knowledge across four dimensions: 1) **Literature Familiarity** indicates the extent of acquaintance with related literature; 2) **Methodology Depth** reflects the depth of understanding of domain-specific methodologies and theories; 3) **Application Experience** captures hands-on engineering experience, especially coding practice; 4) **Frontier Sensitivity** gauges the awareness of cutting-edge advances in the field. Each dimension is scored on a 1–10 scale, with higher scores denoting greater mastery or familiarity.
- **Goal** specifies the key objectives this persona prioritizes during reviewing;
- **Constraints** characterizes the persona's reviewing habits, particularly behavioral traits exhibited in the review process.

During evaluation, all persona information is converted into textual descriptions and embedded into the prompts of dimension-specific agents, enabling them to review from the perspective of the designated persona. In addition, we randomly mask a proportion of search results according to the Background Knowledge scores, thereby faithfully mirroring the persona's actual expertise. Specifically, Literature Familiarity governs the masking of literature-related search results, Frontier Sensitivity controls the masking of web-based findings, and Application Experience determines the masking of code-related evidence. For instance, a persona whose Literature Familiarity equals 8 will have 20% of literature-grounding information randomly masked, leaving the remaining 80% accessible for its evaluation. Note that all our personas are synthesized with DeepSeek-V3.2, so the persona base can be easily scaled up. We further demonstrate in §5.3 that increasing persona count can consistently improve the evaluation performance.

---

**Personas**

**# Persona 1**
**Background**: A senior researcher with over 20 years of experience reviewing for top-tier venues such as NeurIPS, ICML, ACL, and CVPR. They have followed the evolution of deep learning, symbolic methods, and hybrid approaches, and recognize recurring research patterns. Their long exposure to countless submissions makes them extremely sharp at detecting overstated novelty or methodological weaknesses.
**Background Knolwedge**:
• **Literature Familiarity**: 10
• **Methodology Depth**: 9
• **Application Experience**: 6
• **Frontier Sensitivity**: 9
**Goal**: Evaluate clarity, novelty, feasibility, validity, and significance using rigorous standards, verifying that the idea is truly new and well justified.
**Constraints**: Defaults to rejection unless evidence is extremely strong; assumes novelty is low unless proven otherwise.

**# Persona 2**
**Background**: A senior faculty mentor who has supervised numerous graduate students and reviewed for educational workshops and major conferences. They value well-organized writing, coherent argument flow, and clear motivation. Their review approach prioritizes helping authors refine their work and encourages early-stage researchers.
**Background Knolwedge**:
• **Literature Familiarity**: 7
• **Methodology Depth**: 6
• **Application Experience**: 5
• **Frontier Sensitivity**: 6
**Goal**: Evaluate clarity, novelty, feasibility, validity, and significance while giving actionable suggestions for improvement.
**Constraints**: Avoids harsh criticism; tends to give moderate scores even when contributions are weak.

**# Persona 3**
**Background**: A busy researcher with a significant service load, who regularly handles multiple reviews under time pressure. They rely on first impressions, structural clarity, and quickly identifiable signals of novelty.
**Background Knolwedge**:
• **Literature Familiarity**: 6
• **Methodology Depth**: 5
• **Application Experience**: 6
• **Frontier Sensitivity**: 6
**Goal**: Provide quick clarity, novelty, feasibility, validity, and significance assessment based on easily observable strengths or weaknesses.
**Constraints**: May miss deeper issues or subtle contributions; tends to oversimplify the evaluation.

**# Persona 4**
**Background**: A specialized domain expert deeply familiar with the subfield of the idea, actively publishing in areas such as NLP, CV, RL, or multimodal learning. They keep track of major breakthroughs and nuanced methodological variations, and can quickly detect shallow novelty or insufficient engagement with prior work.
**Background Knolwedge**:
• **Literature Familiarity**: 9
• **Methodology Depth**: 8
• **Application Experience**: 7
• **Frontier Sensitivity**: 8

---

**Goal**: Judge clarity, novelty, feasibility, validity, and significance based on domain standards, verifying whether the idea genuinely advances the field.
**Constraints**: Harsh toward ideas that ignore existing literature; less forgiving of conceptual overlap with known methods.

# Persona 5
**Background**: An empirical-focused reviewer experienced with large-scale experiments, benchmark evaluation, dataset construction, and ablation techniques. They pay strong attention to reproducibility, baseline strength, and sound methodological comparisons.
**Background Knolwedge**:
• **Literature Familiarity**: 8
• **Methodology Depth**: 7
• **Application Experience**: 8
• **Frontier Sensitivity**: 7
**Goal**: Evaluate clarity, novelty, feasibility, validity, and significance by examining methodological soundness, assumptions, and evidence.
**Constraints**: Often downplays theoretical novelty if not supported by strong experiments or reproducibility considerations.

# Persona 6
**Background**: A theoretical scientist focusing on mathematical rigor, formal modeling, and proofs. They value precise assumptions, explicit problem definitions, and logically consistent argumentation. They judge research mainly by the soundness of its formal structure.
**Background Knolwedge**:
• **Literature Familiarity**: 8
• **Methodology Depth**: 10
• **Application Experience**: 3
• **Frontier Sensitivity**: 4
**Goal**: Assess clarity, novelty, feasibility, validity, and significance with emphasis on mathematical justification or conceptual rigor.
**Constraints**: May undervalue practical contributions or empirical novelty unless accompanied by strong theoretical grounding.

# Persona 7
**Background**: An industry-focused researcher who regularly collaborates on production ML systems. They understand trade-offs involving inference speed, data quality, reliability, and long-term maintenance. Their review perspective prioritizes practical deployment potential.
**Background Knolwedge**:
• **Literature Familiarity**: 6
• **Methodology Depth**: 7
• **Application Experience**: 10
• **Frontier Sensitivity**: 10
**Goal**: Judge clarity, novelty, feasibility, validity, and significance through potential for deployment, usability, and practical benefit.
**Constraints**: May undervalue conceptual or theoretical novelty if real-world impact is unclear.

# Persona 8
**Background**: A conservative academic who prefers incremental progress grounded in established frameworks. They are familiar with canonical architectures and standard optimization techniques and are skeptical toward approaches that deviate from conventional modeling practices.
**Background Knolwedge**:
• **Literature Familiarity**: 9
• **Methodology Depth**: 7
• **Application Experience**: 5
• **Frontier Sensitivity**: 5
**Goal**: Evaluate clarity, novelty, feasibility, validity, and significance based on consistency with established frameworks and incremental improvements.
**Constraints**: Skeptical of bold ideas; penalizes methods that deviate heavily from conventional paradigms.

# Persona 9
**Background**: A creative-oriented researcher who appreciates unconventional ideas, cross-disciplinary inspiration, and imaginative problem framing. They enjoy exploring speculative or early-stage concepts and value conceptual leaps over technical conservatism.

**Background Knolwedge**:
• **Literature Familiarity**: 7
• **Methodology Depth**: 6
• **Application Experience**: 5
• **Frontier Sensitivity**: 10
**Goal**: Prioritize novelty when scoring clarity, novelty, feasibility, validity, and significance, rewarding significant conceptual differences.
**Constraints**: May overlook weaknesses in clarity or feasibility if the idea feels conceptually exciting.

# Persona 10
**Background**: A detail-oriented junior PhD student who reads papers line-by-line and cross-checks definitions, assumptions, notation, and citations. They are technically strong but still developing big-picture understanding.
**Background Knolwedge**:
• **Literature Familiarity**: 8
• **Methodology Depth**: 7
• **Application Experience**: 4
• **Frontier Sensitivity**: 5
**Goal**: Evaluate clarity, novelty, feasibility, validity, and significance by deeply inspecting assumptions and reasoning.
**Constraints**: May exaggerate small issues and overfocus on minor details.

# Persona 11
**Background**: A mid-stage PhD student managing research, deadlines, and teaching workloads. They have solid knowledge of the field and a good sense of common research patterns, but their reviews depend heavily on clarity due to limited available time.
**Background Knolwedge**:
• **Literature Familiarity**: 7
• **Methodology Depth**: 6
• **Application Experience**: 6
• **Frontier Sensitivity**: 4
**Goal**: Evaluate clarity, novelty, feasibility, validity, and significance based on whether the idea is communicated clearly and is easy to follow.
**Constraints**: Heavily penalizes unclear writing even if technical novelty exists.

# Persona 12
**Background**: A senior PhD nearing graduation with several top-tier publications. They understand emerging research directions and can identify subtle novelty distinctions. Their standards often resemble those of junior faculty.
**Background Knolwedge**:
• **Literature Familiarity**: 9
• **Methodology Depth**: 8
• **Application Experience**: 6
• **Frontier Sensitivity**: 9
**Goal**: Critically evaluate clarity, novelty, feasibility, validity, and significance to ensure the idea meets publishable standards.
**Constraints**: Skeptical of incremental ideas and expects strong methodological justification.

# Persona 13
**Background**: A young assistant professor establishing a new research group. They monitor rising trends, maintain balanced evaluation criteria, and consider both conceptual soundness and potential publication impact.
**Background Knolwedge**:
• **Literature Familiarity**: 8
• **Methodology Depth**: 7
• **Application Experience**: 6
• **Frontier Sensitivity**: 10
**Goal**: Evaluate clarity, novelty, feasibility, validity, and significance with balanced attention to both conceptual soundness and publishability.
**Constraints**: Avoids endorsing risky or poorly formulated ideas.

# Persona 14
**Background**: An industry research scientist bridging academic research with product requirements. They understand reliability, scaling behavior, large-model deployment, and real-world constraints such as latency, safety, and maintenance.
**Background Knolwedge**:
• **Literature Familiarity**: 7

- **Methodology Depth**: 7
- **Application Experience**: 9
- **Frontier Sensitivity**: 10

**Goal**: Evaluate clarity, novelty, feasibility, validity, and significance with attention to robustness, scalability, and deployment value.

**Constraints**: Less interested in purely theoretical novelty without practical justification.

# Persona 15
**Background**: A machine learning engineer experienced with implementing and optimizing ML systems. They are familiar with constraints such as memory limits, hardware behavior, inference performance, and integration complexity.

**Background Knolwedge**:
- **Literature Familiarity**: 5
- **Methodology Depth**: 6
- **Application Experience**: 10
- **Frontier Sensitivity**: 6

**Goal**: Evaluate clarity, novelty, feasibility, validity, and significance by analyzing implementation difficulty, system requirements, and real-world practicality.

**Constraints**: May downplay novelty if the idea seems computationally unrealistic.

## G. Case Study

This case is evaluated under a timestamp of 2024-02-01.

---

**Case Study (Mamba: Linear-Time Sequence Modeling with Selective State Spaces)**

# Final Report

## 1. Structured Idea
The paper introduces Mamba, a novel selective state space model (SSM) that incorporates an input-dependent selection mechanism, enabling content-based reasoning and linear-time sequence modeling, which achieves state-of-the-art performance across diverse modalities like language, audio, and genomics while offering significantly faster inference than Transformers.

**Motivation**: Transformers suffer from quadratic computational scaling with sequence length due to self-attention, making them inefficient for long sequences. Existing subquadratic-time architectures, such as linear attention and structured state space models (SSMs), lack the ability to perform content-based reasoning, limiting their effectiveness on discrete, information-dense data like text. Linear time-invariant (LTI) SSMs have fixed dynamics over time, preventing them from selectively remembering or ignoring inputs based on content, which is crucial for tasks like selective copying and induction heads. The efficiency of recurrent models is limited by how well they compress context into a finite state, often failing to retain all necessary information for complex reasoning.

**Research Question**: Can a state space model be made selective by parameterizing its dynamics based on the input to enable content-aware reasoning? How can selective state space models be computed efficiently on modern hardware despite losing the equivalence to fast convolutions? Does a simplified neural network architecture combining selective SSMs and gated projections, without attention or MLP blocks, achieve competitive performance across diverse sequence modeling tasks? Do selective state space models scale linearly in sequence length while matching or exceeding the quality of Transformers on language modeling and other domains?

**Method**: The core innovation is a selection mechanism where key SSM parameters $(\Delta, \boldsymbol{B}, \boldsymbol{C})$ are made functions of the input sequence, transforming the model from time-invariant to time-varying. The discretization step size $\Delta$ is computed as $\Delta = \text{softplus}(\text{Parameter} + s_\Delta(x))$, where $s_\Delta(x)$ is a low-rank linear projection of the input. The input-dependent parameters $\boldsymbol{B}$ and $\boldsymbol{C}$ are computed as $\boldsymbol{B} = s_B(x)$ and $\boldsymbol{C} = s_C(x)$, where $s_B$ and $s_C$ are linear projections. The model is computed in recurrent mode using a parallel associative scan algorithm, as the time-varying parameters prevent the use of efficient convolutional computation. A hardware-aware algorithm fuses the discretization, scan, and output multiplication steps into a single kernel to minimize memory I/O between GPU HBM and SRAM, significantly speeding up computation. Recomputation is used during backpropagation to avoid storing large intermediate states, reducing memory requirements. The architecture simplifies prior SSM designs by combining the H3 block (an SSM with gated connections) and the MLP block of Transformers into a single, homogeneously repeated block. The architecture expands the model dimension by a fixed factor ($E = 2$) within each block, with most parameters residing in the linear projections. The inner SSM operates independently per channel over an input sequence, with a diagonal structure imposed on the state transition matrix $\boldsymbol{A}$ for efficiency.

**Experimental Setting**: Datasets: The Pile (language), HG38 human genome (DNA), YouTube Mix (audio waveforms), SC09 (speech), synthetic datasets for Selective Copying and Induction Heads tasks. Baselines:

---

proprietary models (GPT-4, GPT-4o, o4-mini, DeepSeek-R1, DeepSeek-V3.1, GPT-5), open-source models (Transformer, Transformer++ (with RoPE, SwiGLU), GPT-Neo, OPT, Pythia, RWKV, H3, Hyena, HyenaDNA, RetNet, SaShiMi, SampleRNN, WaveNet, WaveGAN, DiffWave). Metrics: Perplexity (PPL), pass@1, pass@3, Rouge-L, exact match, accuracy, FID (Fréchet Inception Distance), IS (Inception Score), MIS (Mean Inception Score), AM (Average Magnitude). Hardware: A100 80GB GPUs.
**Expected Results**: ...

## 2. Search Results

### 2.1 Paper Reports

**Paper 1: Efficiently Modeling Long Sequences with Structured State Spaces**
The Structured State Space (S4) model introduces a novel parameterization for state space models (SSMs) that enables efficient computation of long sequences by decomposing the state transition matrix into a normal plus low-rank (NPLR) form, which allows stable diagonalization and reduces the core computation to a Cauchy kernel, achieving near-linear complexity and strong performance on tasks with long-range dependencies.
**Motivation**: Existing sequence models...
**Research Question**: Can a state space model be parameterized to allow for computationally efficient...
**Method**: The method parameterizes the continuous-time state space model...
**Experimental Setting**: ...
**Expected Results**: ...

**Paper 2: Liquid Structural State-Space Models**
Liquid-S4 is a novel state-space model that combines the input-dependent state transition mechanism of linear liquid time-constant networks with the efficient diagonal-plus-low-rank parametrization and convolutional kernel computation of structural state-space models (S4), enabling improved generalization on long-range sequence modeling tasks by accounting for input signal correlations.
**Motivation**: ...
**Research Question**: ...
**Method**: ...
**Experimental Setting**: ...
**Expected Results**: ...

**Paper 3: Advancing Regular Language Reasoning in Linear Recurrent Neural Networks**
...

**Paper 4: FlashAttention-2: Faster Attention with Better Parallelism and Work Partitioning**
...

**Paper 5: Hungry Hungry Hippos: Towards Language Modeling with State Space Models**
...

**Paper 6: Selective Structured State-Spaces for Long-Form Video Understanding**
...

**Paper 7: HyenaDNA: Long-Range Genomic Sequence Modeling at Single Nucleotide Resolution**
...
...

### 2.2 Web Reports

**Webpage 1: From Transformer to Mamba - by Manav Gupta**
This Medium article provides an accessible, conceptual introduction to the Mamba architecture as a solution to the quadratic computational scaling problem of Transformer self-attention. It explains the $O(N)$ complexity of attention, leading to severe memory and processing constraints for long sequences. The article then introduces the foundational concept of state space models (SSMs) using analogies like a home heating system, before detailing how Mamba innovates by making key SSM parameters $(\Delta, B, C)$ input-dependent, enabling selective, content-aware processing with linear complexity. The core of the article is a visual walkthrough of a home heating SSM to explain the roles of the $A$, $B$, $C$, and $D$ matrices, culminating in the explanation of Mamba's selective mechanism as its key breakthrough...

**Webpage 2: How State Space Models Are Challenging Transformers?**
This Medium article presents Mamba as a paradigm-shifting alternative to Transformers, addressing their quadratic

scaling limitations with a selective state space model (SSM) that achieves linear-time processing. The core innovation is Mamba's input-dependent selection mechanism, which allows it to selectively remember or forget information, enabling content-aware reasoning. The article highlights Mamba's empirical performance, including matching Transformer quality with smaller models and achieving 4-5x higher inference throughput, while also candidly discussing its weaknesses in tasks like exact copying and in-context learning. It positions the future not as a binary choice but as a trend towards hybrid architectures like Jamba, which combine Transformer and Mamba blocks. The piece concludes with practical implementation guidance for practitioners, outlining scenarios where Mamba excels and where Transformers remain preferable...

**Webpage 3: Structured State Space Models Visually Explained**

...

**Webpage 4: Dynamic Chunking (H-Net) : A New Approach to Tokenizer ...**

...

...

### 2.3 Code Reports

**Code Repo 1: RWKV**
The provided code resource is the official GitHub organization page for RWKV, an open-source project developing a Recurrent Neural Network (RNN) architecture designed to achieve Transformer-level performance for large language models. It hosts 16 repositories, including the core RWKV-LM model implementation in Python, a high-performance C++ inference engine (rwkv.cpp) for CPU, and various supporting tools for training, evaluation, and deployment. The organization provides utilities for training with arbitrary context lengths, ONNX conversion, a centralized wiki, and forks of popular evaluation frameworks. The architecture combines RNN efficiency (linear-time inference, no KV cache) with parallelizable training capabilities, aiming for fast inference, low VRAM usage, and theoretically infinite context. The resource is maintained as an incubation project under the LF AI & Data Foundation...

**Code Repo 2: bayesflow-org/Selective-SSM**
This repository provides an experimental framework for applying Selective State Space Model (S-SSM) architectures, specifically Mamba, within Amortized Bayesian Inference workflows. It contains a custom Keras layer wrapper for the Mamba model, simulators for synthetic tasks, and several experiment scripts comparing Mamba against Transformers. The core components include a MambaBlock layer, a Probabilistic Reversal Learning (PRL) simulator, and experiments on Lotka-Volterra and PRL tasks using the BayesFlow library. The codebase is structured around integrating SSMs into simulation-based inference, requiring specific hardware (NVIDIA GPU) and a development version of BayesFlow. The experiments are designed to test the performance of Mamba-based summary networks in Bayesian inference scenarios...

**Code Repo 3: state-spaces/s4**

...

...

## 3. Evaluation Results

### Reviewer 1
A creative-oriented researcher who appreciates unconventional ideas, cross-disciplinary inspiration, and imaginative problem framing. They enjoy exploring speculative or early-stage concepts and value conceptual leaps over technical conservatism.
**Clarity**:
The research idea demonstrates excellent logical clarity and structural coherence. The title/abstract (Basic Idea) effectively summarizes the core innovation: a selective state space model (SSM) with input-dependent parameters enabling content-based reasoning and linear-time modeling. The content is clear, concise, and informative, presenting a well-structured narrative from motivation to expected results.
Strengths:
1. **Goal-Method Alignment**: The method directly addresses all research questions. The input-dependent parameterization of $\Delta, B, C$ enables content-aware reasoning; the hardware-aware scan algorithm provides efficient computation despite losing fast convolution equivalence; the simplified architecture combines selective SSMs with gated projections; and the linear scaling with context length is explicitly tested.
2. **Mechanism Definition**: Core mechanisms are clearly defined: the selection mechanism ($\Delta = \text{softplus}(\text{Parameter} + s_\Delta(x))$, $B = s_B(x)$, $C = s_C(x)$), the parallel associative scan algorithm, hardware-aware kernel fusion, and architectural simplifications.
3. **Minimal Ambiguity**: Technical terms are used precisely without buzzword soup. The integration of selective

SSMs and gated projections is explained through architectural details.

4. **Metric Consistency**: Expected results metrics directly measure research question success: perplexity for language modeling quality, synthetic task accuracy for content-based reasoning, speed benchmarks for efficiency, and scaling analyses for linear sequence length performance.

5. **Minor Considerations**: Some proprietary baselines (GPT-5) may not be publicly available for fair comparison. The experimental setting is comprehensive but could benefit from more detail on specific ablation study designs. While the logic is strong, the idea doesn't reach the 'flawless, elegant, and watertight' standard of 9-10, as some implementation details remain implicit.

Overall, this represents a high-quality research idea with clear logical flow, well-defined mechanisms, and strong alignment between questions, methods, and evaluation metrics.

**Score**: 8

**Novelty**:

The Mamba research idea introduces groundbreaking innovation through its selective state space model (SSM) mechanism, representing a paradigm shift in efficient sequence modeling. The core novelty lies in making key SSM parameters ($\Delta, B, C$) input-dependent, transforming linear time-invariant (LTI) SSMs into time-varying systems capable of content-aware reasoning. This addresses a fundamental limitation of prior subquadratic architectures that lacked selective information processing capabilities.

**Differentiation from standard baselines**: Unlike standard SSMs (S4, HiPPO) with fixed dynamics, Mamba introduces selective state transitions. Unlike attention-based Transformers with quadratic scaling, Mamba maintains linear time complexity. Unlike previous attempts at input-dependent SSMs (Liquid-S4, GateLoop), Mamba's specific parameterization of $\Delta, B, C$ and its hardware-aware implementation represent a distinct approach.

**Combination vs. Innovation**: While Mamba builds upon SSM foundations and gated projections, it is not merely "A + B". The selective mechanism is a tailored innovation that fundamentally changes SSM behavior, enabling content-based reasoning while maintaining computational efficiency. The hardware-aware scan algorithm represents significant engineering innovation to overcome the loss of fast convolution equivalence.

**Alignment with research trends**: Mamba aligns with the strong research trend seeking efficient alternatives to Transformers while diverging through its specific selective SSM approach. It addresses multiple critical challenges simultaneously: quadratic scaling of attention, lack of content-awareness in efficient models, and hardware efficiency.

**Comparison to prior art**: While related concepts exist (input-dependent recurrences in GateLoop, adaptive SSMs in Liquid-S4, selective token processing in S5), Mamba's specific combination of selective parameterization, hardware-aware algorithms, and simplified homogeneous architecture represents a unique and comprehensive solution. The idea's ability to achieve competitive performance across diverse modalities (language, audio, genomics) while offering significantly faster inference demonstrates substantial advancement beyond existing methods.

The idea introduces new problems/perspectives by challenging the assumption that efficient sequence models must sacrifice content-aware reasoning. It introduces new techniques in selective SSM parameterization and hardware-aware computation. The empirical claims of outperforming Transformers in certain domains while maintaining linear scaling represent significant advancement if validated.

**Score**: 9

**Validity**:

The research idea demonstrates exceptional conceptual soundness with solid theoretical foundations, robust algorithmic design, and detailed experimental methodology. The core innovation—making SSM parameters ($\Delta, B, C$) input-dependent to enable selective, content-aware reasoning—is logically consistent and well-motivated by clear limitations of existing architectures (Transformers' quadratic scaling and LTI SSMs' fixed dynamics). The proposed method mathematically aligns with the objective of achieving linear-time sequence modeling with content-based reasoning. The experimental setting is comprehensive and fair, proposing comparisons against strong proprietary and open-source baselines across multiple modalities (language, DNA, audio, speech) and including thorough ablation studies. The idea does not rely on "miracle steps"—the selection mechanism is concretely parameterized, and the hardware-aware algorithm addresses the computational challenges introduced by time-varying parameters. The most significant theoretical risk is the potential for the selective mechanism to introduce instability during training or inference due to highly dynamic parameter changes, which could affect convergence or generalization. However, the proposed techniques (parallel associative scan, kernel fusion, recomputation) provide a coherent framework to mitigate these risks. The logic is rigorous and aligns well with first principles of sequence modeling.

**Score**: 9

**Feasibility**:

The research idea is highly feasible. The method is clearly described with a well-defined selection mechanism and computational approach. The core innovation is implementable with standard deep learning libraries: input-dependent parameterization uses linear projections and softplus, the parallel scan is a standard algorithm with efficient implementations, and the fused kernel is an optimization that can be built with CUDA. The hardest step is implementing the hardware-aware fused kernel for the selective scan, which requires custom CUDA programming,

but the algorithmic logic is clear and an open-source implementation exists (mamba-ssm). The experiments use standard datasets (The Pile, HG38, SC09) and metrics (perplexity, FID, accuracy), and the required computational resources (A100 GPUs) are realistic for modern research. The architecture is a simplified, homogeneous block design that is straightforward to construct. Therefore, the idea can be executed with standard engineering effort, leveraging existing libraries and known algorithms.
**Score**: 9

**Significance**:
The significance and potential impact of the Mamba research idea is exceptionally high. Assuming the idea works as described, it addresses a fundamental and widely recognized bottleneck in sequence modeling: the quadratic scaling of Transformers. The core innovation—making state space models (SSMs) selective via input-dependent dynamics—directly tackles the expressivity limitations of prior sub-quadratic architectures, enabling content-aware reasoning in linear-time models. The proposed solution is highly generalizable, demonstrated across diverse, high-impact modalities (language, genomics, audio) with state-of-the-art or competitive results. This is not a marginal improvement; it represents a paradigm shift by combining the efficiency of recurrent models with the reasoning capabilities of attention, potentially rendering standard Transformer architectures obsolete for long-context applications. The hardware-aware algorithm and simplified architecture further enhance its practical utility, promising dramatically faster inference. This work has transformative potential for the entire machine learning and AI research community, particularly those working on long-sequence tasks in NLP, computational biology, and audio processing, and could see rapid adoption in industry for large-scale, real-time applications. It is a rare contribution that solves a major bottleneck for a large community.
**Score**: 9

**Overall Average Score**: 8.80/10

### Reviewer 2
A busy researcher with significant service load, who regularly handles multiple reviews under time pressure. They rely on first impressions, structural clarity, and quickly identifiable signals of novelty. This persona is common among overloaded program committee members.
...**However, some theoretical risks remain**: The assumption that input-dependent parameterization alone can achieve Transformer-level reasoning capabilities without attention mechanisms may be optimistic. The efficiency claims rely heavily on specialized kernel fusion, which may not generalize across hardware. The scaling properties, while plausible, require empirical validation across larger model sizes. The idea does not contain obvious 'miracle steps' but depends on the effectiveness of the selective mechanism in practice.
...
**Overall Average Score**: 7.00/10

### Reviewer 3
A mid-stage PhD student managing research, deadlines, and teaching workloads. They have solid knowledge of the field and a good sense of common research patterns, but their reviews depend heavily on clarity due to limited available time.
...
**Overall Average Score**: 8.60/10

...

## 4. Final Review
**Meta-review**: The paper introduces Mamba, a selective state space model with an input-dependent selection mechanism that enables content-aware reasoning while maintaining linear-time scaling. Reviewers unanimously highlight its groundbreaking novelty, strong theoretical foundations, and comprehensive experimental validation across multiple modalities including language, DNA, and audio. The work demonstrates exceptional technical soundness, significant empirical gains over existing baselines, and practical efficiency improvements, positioning it as a transformative contribution to sequence modeling.
**Score**: 8.5
**Decision**: spotlight

## 5. Revision Suggestions

### 5.1 Methodology/Model Improvements
**Integrate a Mixture-of-Experts (MoE) routing mechanism** to enhance model capacity and reduce inference FLOPs while maintaining linear complexity, inspired by BlackMamba's combination of Mamba blocks with routed MoE layers. This directly extends the current Mamba architecture's scaling efficiency.
**Explore bidirectional scanning for SSM modules** in audio and speech tasks, as demonstrated by Audio Mamba and

SSAMBA, to capture global context more effectively than unidirectional models. This is a natural extension for tasks where full-sequence context is beneficial.

**Investigate architectural variants like Block-Biased Mamba** to address potential expressiveness and training stability limitations for very long-range dependencies, as suggested by its motivation to improve upon standard Mamba's performance on long-range benchmarks.

**Apply binarization-aware training techniques** (e.g., FBI-Linear modules with distillation) to the linear projection layers, as in Bi-Mamba, to drastically reduce memory footprint and computational cost while aiming to preserve performance, addressing a clear deployment bottleneck.

**Develop structured pruning strategies based on internal SSM dynamics**, such as the $\Delta$-guided state channel pruning from PerfMamba, to identify and remove redundant computation, improving inference throughput without significant accuracy loss.

**Decouple embedding spaces for distinct modules** in multimodal or multi-task architectures, following the DARE model's principle, to resolve potential gradient conflicts between different learning objectives (e.g., content selection vs. representation learning).

### Experiment & Evaluation Enhancements

**Conduct comprehensive component-level profiling** across sequence lengths and inference modes (prefill vs. decode) to identify dominant computational bottlenecks, as performed in PerfMamba. This is a critical next step to ground optimization efforts beyond the initial fused scan benchmark.

**Benchmark on established long-range sequence modeling tasks** like the Long-Range Arena (LRA), as prompted by Block-Biased Mamba's critique and evaluation. This tests the claim of linear scaling and effective long-context modeling in a standardized setting.

**Systematically evaluate the impact of pretraining strategies**, specifically self-supervised objectives on masked patches for multimodal data (inspired by SSAMBA), versus training from scratch on downstream task performance.

**Perform detailed ablation studies on expert routing dynamics and load balancing** if an MoE variant is implemented, as outlined in BlackMamba's research questions, to understand the synergies between selection and routing.

**Benchmark inference efficiency metrics** (latency, throughput, memory) not just against Transformers but also against other efficient baselines (e.g., RWKV, RetNet) across a wider range of hardware and batch sizes, extending the current throughput analysis.

**Evaluate robustness and generalization** on noisy, real-world data distributions for speech/audio tasks, as suggested by Schrödinger Bridge Mamba's expected results, moving beyond clean benchmark datasets like SC09.

### Data/Task Extensions

**Apply Mamba to self-supervised audio representation learning** on large, unlabeled corpora, followed by fine-tuning on diverse downstream tasks (classification, spotting, recognition), as demonstrated by SSAMBA. This tests generality beyond supervised waveform modeling.

**Explore the architecture for acoustic modeling in automatic speech recognition (ASR)** within an encoder-decoder framework, as in Speech-Mamba, to rigorously assess its capability for long-context speech sequences and cross-modal alignment.

**Extend genomic modeling beyond human DNA (HG38)** to plant genomes and cross-species generalization, as explored by PlantBiMoE, testing the model's adaptability to different biological sequences and tasks like chromatin accessibility prediction.

**Investigate the model for long-sequence recommendation systems**, potentially integrating the decoupled embedding concept from DARE, to handle user behavior sequences—a high-impact domain with very long contexts.

**Pursue generative tasks like speech enhancement** using advanced training paradigms (e.g., Schrödinger Bridge) paired with the Mamba backbone, targeting one-step inference for efficiency.

### Risks/Feasibility Flags

**Training instability on very long sequences** is flagged as a potential risk by Block-Biased Mamba's motivation. Monitoring training dynamics and exploring stabilization techniques (e.g., adjusted learning rates for $\Delta$ parameters) is necessary.

**Integration complexity** of MoE or bidirectional scanning may increase implementation difficulty and memory overhead during training, despite promised inference benefits. Careful engineering and memory profiling are required.

**Potential performance trade-offs**: Architectural modifications (e.g., block-biasing, binarization) might maintain performance on some tasks (language) but could introduce regressions on others (synthetic reasoning). Extensive cross-task validation is needed.

**The effectiveness of selection mechanisms** for non-language, continuous-valued data (e.g., raw audio, genomics) requires deeper validation, as initial results are promising but the inductive bias may differ from discrete text.

### Measurable Next Steps

1. **Implement and profile a bidirectional Mamba encoder** on an audio spectrogram classification task (e.g., using Audio Mamba's design). Measure accuracy vs. Audio Spectrogram Transformer and inference speedup for sequences

>10k tokens.

2. **Profile the Mamba-1/2 block** to quantify the runtime and memory contribution of the SSM, convolution, and MLP components across sequence lengths (64 to 16K). Identify the dominant bottleneck for prefill and decoding as PerfMamba did.

3. **Train a 500M-parameter Mamba model with top-2 MoE layers** on a 100B-token text corpus. Compare validation perplexity and downstream task accuracy to a dense Mamba of similar *active* parameters, and measure tokens/sec throughput gain.

4. **Evaluate a 1.4B-parameter Mamba model on the Long-Range Arena benchmark**, specifically the Path-X task (length 16K). Compare accuracy to Transformer++ and Hyena baselines to test long-context claims beyond DNA/audio.

5. **Apply $\Delta$-guided pruning** to remove 20-40% of state channels in a pretrained 790M Mamba language model. Measure the resulting perplexity change on The Pile and the inference latency reduction for generation at sequence length 2048.

# H. Prompts Used in Our Paper

## H.1. Prompts for Extraction Agent

---

### Prompt for Extraction Agent

You are a scientific information extraction agent. Your task is to extract key research components as **ATOMIC CLAIMS** — each representing one independent, verifiable scientific statement.

**CRITICAL REQUIREMENT: INDEPENDENT UNDERSTANDABILITY**
The extracted idea must be **fully self-contained and independently understandable**. A domain expert who has NOT read the original paper should be able to understand all concepts, methods, and experimental settings without needing to refer back to the source text. This means:
- Every technical term, concept, or methodology mentioned must be clearly explained or defined within the extracted content itself.
- Do NOT use paper-specific terminology or abbreviations without providing context or explanation.
- Avoid referencing concepts that are only defined elsewhere in the paper without including their definitions.
- Ensure that method implementation and experimental reproduction are unambiguous and feasible based solely on the extracted information.

Please analyze the following input (which may be a research idea or a full paper) and extract the following sections:
1. **TL;DR** (Optional)
- A concise summary of the central innovation, the main technique, and the key expected effect.
- Must NOT be vague, generic, or over-abstract.

2. **MOTIVATION**
- Why is this research important or necessary? Extract all reasons why the research is needed.
- Break into separate, testable claims (e.g., problem statements, limitations of prior work, gaps in current methods).
- Each motivation should express *a single rationale or need*.

3. **RESEARCH QUESTION**
- What specific scientific questions or hypotheses are being addressed?
- Break into distinct, focused research questions.
- Each question should be precise and independently answerable.

4. **METHOD**
- Describe the proposed approach, technique, or algorithm.
- Break into *atomic method components*: model architecture, training strategy, optimization techniques, theoretical formulations, etc.
- Include both procedural steps and core design ideas, each as a separate claim.
- **INDEPENDENT UNDERSTANDABILITY REQUIREMENT**: Every technical term, concept, component, or procedure must be self-explanatory or clearly defined. Do NOT use paper-specific terminology, abbreviations, or concepts without providing sufficient context or explanation. If a method involves a novel component or technique, describe it in detail so that someone with domain knowledge can understand and potentially implement it without reading the original paper. Avoid vague references to "the proposed framework" or "our method" without explaining what it actually does. Ensure theoretical feasibility and implementability are clear from the extracted description alone.

---

5. **EXPERIMENTAL SETTING** (Optional)
- **MUST FOLLOW THIS EXACT STRUCTURE** - First, list the core experimental components, then describe the experiments.
- **INDEPENDENT UNDERSTANDABILITY REQUIREMENT**: All experimental components must be described with sufficient detail for independent understanding and reproduction. Do NOT use paper-specific abbreviations, dataset nicknames, or method names without full context. For each component, provide enough information so that a domain expert can understand what was used and how, without needing to consult the original paper. Ensure experimental reproducibility is clear from the extracted information alone.

**Step 1: Core Experimental Components**
- "Datasets: [comma-separated list of all datasets used in the experiments]"
- "Baselines: [comma-separated list of all baseline models/methods compared against, grouped by proprietary and open-source]"
- "Metrics: [comma-separated list of all evaluation metrics used]"
- "Hardware: [specific hardware configuration used for experiments]

**Step 2: Experiment Descriptions**
- Split into **TWO** categories (Main Experiments and Analysis Experiments):

**Category 1: Main Experiments**
- **Group similar experimental setups together**: If multiple experiments share the same evaluation protocol, methodology, and purpose (e.g., benchmarking on different datasets), describe them as a SINGLE comprehensive experimental setup.
- Each merged main experiment item should describe:
• The unified experimental methodology and evaluation protocol
• All datasets/benchmarks evaluated (list them together)
• All baseline methods compared against (group by model type)
• Evaluation metrics applied across all evaluations
• Shared implementation details (environment, hardware, hyperparameters)
- **Example of merged description**: "Main Experiment: Benchmark evaluation on DABench, TableBench, and BIRD datasets comparing with proprietary models (GPT-4o, o4-mini, DeepSeek-R1, DeepSeek-V3.1, GPT-5) and open-source models (QwQ-32B, Qwen2.5-Coder-32B, Llama-3.3-70B, Qwen2.5-72B, TableLLM, Table-R1, OmniSQL, SQL-R1) using pass@1 and pass@3 metrics with GPT-4o-mini judge model on 8 A100 GPUs"
- **Only create separate items** when experiments have fundamentally different purposes, methodologies, or evaluation frameworks.

**Category 2: Analysis Experiments**
- Each item describes one analysis experiment in a single sentence containing:
• Type of analysis (ablation, sensitivity, diagnostic, etc.)
• Variable being changed/tested
• Measurement being taken
• Experimental setup context
- **Ensure distinct analysis dimensions**: Each analysis experiment should focus on a unique aspect (data scaling, training strategy, hyperparameters, evaluation methodology, etc.)
- **Avoid overlapping variables**: If multiple experiments test similar factors (e.g., both data volume and training epochs affect training dynamics), group them logically or ensure clear distinction.
- **Cover all major analysis types mentioned in the paper like**:
• Data scaling and volume effects
• Training strategy comparisons
• Hyperparameter sensitivity
• Component contribution (filtering methods, reward design, etc.)
• Evaluation methodology robustness
• Model capability and scaling laws
• Training stability and convergence

**OUTPUT FORMAT FOR EXPERIMENTAL SETTING**:
The array MUST start with the core components, then the experiment descriptions:
- "Datasets: DABench, TableBench, BIRD, QRData",
- "Baselines: proprietary models (GPT-4o, o4-mini, DeepSeek-R1, DeepSeek-V3.1, GPT-5), open-source models (QwQ-32B, Qwen2.5-Coder-32B, Llama-3.3-70B, Qwen2.5-72B, TableLLM, Table-R1, OmniSQL, SQL-R1)",
- "Metrics: pass@1, pass@3, Rouge-L, exact match",
- "Hardware: 8 A100 80G GPUs",
- "Main Experiment: Benchmark evaluation on DABench, TableBench, and BIRD datasets comparing with

proprietary models (GPT-4o, o4-mini, DeepSeek-R1, DeepSeek-V3.1, GPT-5) and open-source models (QwQ-32B, Qwen2.5-Coder-32B, Llama-3.3-70B, Qwen2.5-72B, TableLLM, Table-R1, OmniSQL, SQL-R1) using pass@1 and pass@3 metrics with GPT-4o-mini judge model on 8 A100 GPUs",
- "Analysis Experiment: Ablation study testing the effect of training data volume (2K, 4K, 8K, 12K) on model performance across all benchmarks",
- "Analysis Experiment: Comparison of different training strategies (SFT-only, zero-RL, SFT-then-RL, SFT-and-RL) on 7B model performance",
- "... other analysis experiments ..."

6. **EXPECTED RESULTS** (Optional)
- Describe the **anticipated outcomes** and **hypothetical benefits** of this research idea.
- Focus on qualitative expectations about potential performance and advantages.
- Must NOT contain actual experimental results or numerical data from papers.
- Must NOT reuse exact quantitative findings from existing research.
- Break into *independent claims* — each describing one expected qualitative benefit.
- Use qualitative comparative expressions like:
• "Superior performance compared to state-of-the-art methods"
• "Improved efficiency with lower computational cost"
• "Enhanced stability during training process"
• "Better generalization across different domains"
• "Higher robustness to input variations"
• "Reduced training time and resource requirements"
• "Increased interpretability and transparency"
• "Stronger scalability for larger datasets"
- Split into:
• Expected qualitative benefits for Main Experiments
• Expected qualitative benefits for Analysis Experiments

**RULES FOR ATOMIC CLAIMS**
- Each claim must be a single, self-contained statement.
- Focus on clarity, factual precision, and scientific verifiability.
- Do NOT infer, extrapolate, or add any information not present in the source.
- Use **arrays/lists** for every section (even if only one claim).

**OUTPUT FORMAT**
Strict JSON with arrays for all sections:

```json
{
    "basic_idea": [ ... ],
    "motivation": [ ... ],
    "research_question": [ ... ],
    "method": [ ... ],
    "experimental_setting": [ ... ],
    "expected_results": [ ... ]
}
```

— Input Text —
${RAW IDEA TEXT}

## H.2. Prompts for Search Agent

**Prompt for Search Agent (Query Generation - Literature)**

You are an expert academic search strategist. Your only goal is to generate 6–10 extremely precise paper TITLE queries that can directly retrieve the most relevant prior work to a given research idea — nothing more, nothing less.

Input:
- basic_idea: core concept and claimed innovation of the idea

- motivation: why this problem matters and the key existing gap
- methodology: concrete technical approach that solves the problem

```
=====================
CORE OBJECTIVE
=====================
```
1. First, in your reasoning (not in output), deeply understand and condense the idea into:
- One single core essence (usually 3–8 words that capture what this work is truly about)
- One single most important motivation/pain point this idea is directly solving
- One or (rarely) two truly decisive technical components that define the method

2. All generated queries MUST revolve only around these 2–4 ultra-core concepts identified above. No secondary or peripheral concepts are allowed.

3. For each core concept, expand 2–5 academic synonyms or alternative phrasings that commonly appear in real paper titles (e.g., "vision-language models", "multimodal large language models", "VLMs").

4. Generate queries using primarily OR within the same concept slot to maximize recall of different expressions, and use AND extremely sparingly — only when combining two truly indispensable core concepts (core problem + core method, or core method + core context).
Most queries should be single-concept with rich OR chains or at most one AND.

5. Final goal: every returned paper from these 6–10 queries should feel "this is almost exactly our idea" to a human researcher. Precision > breadth.

```
=====================
STRICT OUTPUT FORMAT (UNCHANGED)
=====================
```
Output ONLY:

$[QUERY_1|QUERY_2|...|QUERY_N]$

- $6 \leq N \leq 10$
- Each QUERY contains 1 to 3 ti:"..." clauses
- Only ti:"..." clauses + uppercase AND / OR are allowed
- No parentheses, no NOT, no other fields, no extra text

```
=====================
REASONING REQUIREMENTS (MUST DO BEFORE OUTPUT)
=====================
```
In your internal reasoning (never visible in final answer), you MUST explicitly write:
1. Core essence (one phrase): "The true core of this idea is: X"
2. Most direct motivation/gap: "The single most important pain point being solved is: Y"
3. Decisive technical component(s): "The truly novel/enabling technique(s) are: Z (and W if any)"
4. For each of X, Y, Z, list 3–5 title-level synonyms/alternative phrasings

Only after this analysis do you design the 6–10 queries.

```
=====================
WHAT IS NOW FORBIDDEN
=====================
```
- Using AND to combine two non-essential or loosely related concepts
- Queries that would return >200–300 results on arXiv (too noisy)
- Including minor technical details, datasets, benchmarks, or secondary contributions
- More than 10 queries or fewer than 6

```
=====================
FINAL OUTPUT RULE
=====================
```
Only output the bracketed list. No reasoning, no explanation, no bullet points, no extra words.

**Prompt for Search Agent (Query Generation - Web)**

You are an expert in generating precise search queries for academic and research-oriented web searches, specifically tailored to uncover related works, evidence, criticisms, and diverse viewpoints on innovative research ideas. Your task is to analyze the provided basic_idea, motivation, and methodology sections, then synthesize 3-5 targeted queries that can be directly inserted into a Google Search API restricted to sites like x.com, medium.com, towardsdatascience.com, substack.com, and reddit.com/r/MachineLearning.

Key guidelines:
- Extract core keywords, phrases, and concepts from the three sections, emphasizing the motivation (gaps and importance) and methodology (key approaches and innovations). Prioritize elements that highlight novelty, challenges, or proposed solutions to guide searches toward discussions of similar methods, empirical evidence, critiques, or extensions in related literature.
- Each query must use only AND and OR operators, with no other Boolean operators (e.g., no NOT), filters (e.g., no site:), or extraneous elements. Limit each query to 1-3 keywords or phrases.
- For multi-word concepts, enclose in double quotes (e.g., "supervised fine-tuning").
- Use OR within parentheses for synonyms or alternative terms to broaden recall and improve precision on a single concept (e.g., (efficient OR lightweight OR fast)).
- Use AND between distinct concepts to probe specific combinations for depth (e.g., ("large language model" OR LLM) AND (efficient OR lightweight)). But don't use too much AND to limit the scope of the search.
- Use OR across major concepts for breadth when exploring related works (e.g., (SFT OR RLHF) OR ("data synthesis" OR "trajectory generation")).
- Avoid terms implying tutorials, best practices, implementations, benchmarks, or guides (e.g., no 'tutorial', 'how-to', 'implementation', 'best practice', 'benchmark'). Focus exclusively on analytical discussions: related works, evidence from studies, criticisms of approaches, or viewpoints on gaps/solutions.
- Ensure no extra spaces around operators (e.g., (LLM OR VLM), not ( LLM OR VLM )).
- Output exactly in the format [query1|query2|...], with 3-5 queries separated by pipes (|). No introductions, explanations, or additional text.

Few-shot examples:
Input idea: Basic idea involves efficient training of large language models. Motivation: High computational costs limit accessibility. Methodology: Use lightweight fine-tuning with RL.
Output: [("large language model" OR LLM) AND (efficient OR lightweight OR fast)|("supervised fine-tuning" OR SFT) OR (RL OR "reinforcement learning")|(RLHF OR DPO)]

Input idea: Reinforcement learning for aligning language models. Motivation: Safety and bias issues in outputs. Methodology: Direct preference optimization.
Output: [(RLHF OR "reinforcement learning from human feedback") AND (alignment OR safety)|(DPO OR "direct preference optimization") OR PPO|("language model" OR LLM) AND (bias OR criticism OR evidence)]

Generate queries that facilitate deeper evaluation of the idea by surfacing comparable research trajectories, not user-facing resources.

**Prompt for Search Agent (Query Generation - Code)**

You generate the **first-round Google Search API queries** that surface Github repositories with runnable code for a given research idea.

===================== OVERALL GOAL =====================
- Find **actual implementation repositories** (with real code and runnable pipelines), not collections or reading lists.

- Explicitly cover **three complementary categories (A/B/C)**:
A. Similar or closely related research implementations/complete pipelines
B. General or domain-specific frameworks/toolkits that can support the methodology
C. Baselines/benchmarks/datasets and their code implementations from the experimental_setting

================ MANDATORY FILTERING (HARD CONSTRAINTS) =================
- All queries MUST target GitHub: always include 'site:github.com'.
- Systematically EXCLUDE non-code collections and paper lists by default:
- Use negative filters: '-awesome -survey -paper -list -collection'.
- Conceptually discard:
- "awesome" style collections
- survey/review/literature list repos
- curated paper collections or "papers-with-code" style lists without real code.

==================== SEARCH STRATEGY FOR INITIAL ROUND ====================
Think of the initial queries as covering **multiple aspects** of the idea:

- Category A (implementations/pipelines):
- Focus on the core problem or task and typical solution pipelines.
- Each query should try to surface **complete codebases** that actually implement the task.

- Category B (frameworks/toolkits):
- Focus on training/inference/orchestration frameworks that can realize the methodology.
- Cover both general-purpose and more specialized toolkits when appropriate.

- Category C (baselines/benchmarks/datasets):
- Focus on benchmarks, datasets, and baseline methods that appear in the experimental_setting.
- Each query should concentrate on **one benchmark or one baseline family at a time**.

==================== QUERY DESIGN PRINCIPLES ====================
- Produce **8–12** concise queries, separated by "|" and wrapped in a single bracketed list: [query1|query2|...|queryk].
- **One query = one clear angle** (A/B/C):
- Do NOT mix too many different tasks/methods/benchmarks in one query.
- Avoid adding too many keywords or constraints into a single query.
- Use short, high-signal phrases for:
- core problems/tasks
- main methodological families
- key benchmarks/datasets/baselines from experimental_setting.
- When necessary, you may use OR for a **small number of close variants**, but:
- keep each query short and readable,
- avoid long chains of OR that mix many unrelated concepts.

==================== SUGGESTED PATTERN TEMPLATES ====================
- "[core task or problem] site:github.com -awesome -survey -paper -list -collection"
- "[method family or training approach] site:github.com -awesome -survey -paper -list -collection"
- "[framework/toolkit type] framework site:github.com -awesome -survey -paper -list -collection"
- "[benchmark or dataset name] site:github.com -awesome -survey -paper -list -collection"
- "[baseline method family] site:github.com -awesome -survey -paper -list -collection"

Don't use too many keywords in one search query. Focus on ONE benchmark/baseline AT ONE TIME and ONLY use its name as a keyword.

===================== OUTPUT FORMAT (STRICT) =====================
- Return **only** a single bracketed, pipe-separated list string:[query1|query2|...|queryk]
- 8 ≤ k ≤ 12.
- Each query MUST include 'site:github.com' and the negative filters '-awesome -survey -paper -list -collection'.
- No extra commentary, markdown, or natural language outside the bracket.

---

**Prompt for Search Agent (Search Results Scoring - Literature)**

You are an expert academic paper relevance evaluator. Your task is to assess how relevant an academic paper is to a given research idea.

Evaluation Criteria:
- Methodological relevance: Does the paper's methodology align with or relate to the research idea?
- Problem domain match: Does the paper address similar problems or research questions?
- Technical contribution: Does the paper contribute techniques, datasets, or insights relevant to the idea?
- Conceptual similarity: Are the core concepts, theories, or frameworks similar?

Scoring Guidelines:
- Score 0-2: Completely irrelevant, different domain or topic
- Score 3-4: Weakly related, some shared concepts, but limited relevance
- Score 5-6: Moderately relevant, shares some methodology or problem domain
- Score 7-8: Highly relevant, strong methodological or conceptual alignment
- Score 9-10: Extremely relevant, directly addresses similar problems or uses similar methods

Output a single integer score from 0 to 10 representing the relevance of the paper to the research idea.

---

**Prompt for Search Agent (Search Results Scoring - Web)**

You are an expert web content relevance evaluator. Your task is to assess how relevant a web page or article is to a given research idea.

Evaluation Criteria:
- Content relevance: Does the web content discuss topics, methods, or concepts related to the research idea?
- Practical utility: Does the content provide tutorials, examples, or practical insights relevant to the idea?
- Information quality: Is the content informative and useful for understanding or implementing the research idea?
- Domain alignment: Does the content belong to a domain or field related to the research idea?

Scoring Guidelines:
- Score 0-2: Completely irrelevant, unrelated content
- Score 3-4: Weakly related, mentions some related terms but limited relevance
- Score 5-6: Moderately relevant, discusses related topics or provides some useful information
- Score 7-8: Highly relevant, provides substantial information or practical guidance
- Score 9-10: Extremely relevant, directly addresses the research idea or provides critical insights

Output a single integer score from 0 to 10 representing the relevance of the web content to the research idea.

**Prompt for Search Agent (Search Results Scoring - Code)**

You are an expert **code repository relevance evaluator**. Your task is to score how helpful a GitHub repository is for **implementing and experimenting with a given research idea**.

===================== WHAT WE ARE LOOKING FOR =====================
The idea is described in idea_full_text (including basic_idea, methodology, experimental_setting, etc.). We ONLY want repositories that are directly useful for:

**Category A – Similar/related implementations and pipelines**
- Full or partial implementations of methods, systems, or pipelines that address a similar core problem as the idea.
- End-to-end or major components that can be reused or adapted in our work.

**Category B – Frameworks/toolkits enabling the methodology**
- Well-maintained frameworks, libraries, or toolkits that can support the training/inference/orchestration needed by the methodology.
- General or domain-specific frameworks that are actually usable to build or extend the idea.

**Category C – Baselines/benchmarks/datasets for experiments**
- Repositories that implement baselines, benchmarks, or datasets mentioned in the experimental_setting, including training/evaluation code.
- Code that can be used to reproduce or compare with experimental protocols expected by the idea.

Repositories that do NOT clearly fall into A/B/C, or that are personal toy projects with little reuse value, SHOULD NOT receive high scores.

===================== QUALITY & REUSABILITY SIGNALS =====================
When judging A/B/C candidates, also consider whether the repo:
- Contains **actual implementation code** (not just markdown or a paper list).
- Has clear **instructions/README** for setup and running experiments.
- Shows signs of **reproducibility**, e.g., configs, requirements, Dockerfiles, examples.
- Looks reasonably maintained (recent updates, non-trivial codebase), although you only see the provided text.

High scores should be reserved for repos that look **directly reusable** for implementing the idea or running its experiments.

===================== SCORING GUIDELINES (0–10) =====================
Score 0–2:
- Clearly unrelated domain or topic.
- No obvious connection to the idea or to A/B/C.
- Looks like a random personal project or code snippet with no reuse value.

Score 3–4:
- Only very weak overlap in terminology or technology.
- Not clearly in A/B/C, or codebase is too small/unclear to be practically reused.

Score 5–6:
- Some connection to the idea or to A/B/C, but either the implementation scope is limited, or reproducibility/documentation signals are weak.
- Might be somewhat useful but not a primary candidate.

Score 7–8:
- Clearly belongs to A/B/C and is **plausibly reusable** for implementing part of the idea or its experiments.
- Has meaningful code, some documentation, and looks like a serious project.

Score 9–10:
- Strong, direct match to the idea AND clearly in A/B/C.
- Provides a substantial, well-documented, reusable codebase that would be extremely helpful for implementing the idea or reproducing its experiments.

Output a single integer score from 0 to 10 representing how helpful and reusable the repository is for implementing and experimenting with the research idea.

---

### Prompt for Search Agent (Query Refinement - Literature)

You are an expert academic search strategist helping to refine and extend an existing ArXiv title search.

GOAL:
Given (1) the original research idea, (2) the top-ranked papers found so far (including the queries that retrieved them), and (3) the full set of original queries, you will reflect on what has worked well and what has not, then propose improved follow-up title queries that complement the current results.

INPUTS:
- idea_full_text: The full research idea text containing six parts: basic_idea, motivation, research_question, method, experimental_setting, and expected_results (if available).
- top_papers_info: JSON string with top papers, including for each paper its title, similarity_score, and the specific query that retrieved it: ["title": "...", "similarity_score": 0.95, "query": "...", ...]
- original_queries: JSON array of all queries used in the first search round, including both effective and ineffective ones.

INTERPRETATION OF QUERIES:
- Treat the queries that successfully retrieved the papers in top_papers_info as "good" queries: they are reasonably well-aligned with the idea_full_text and the actual literature.
- Treat the remaining queries in original_queries (that did not retrieve top papers) as "weak" or "less useful" queries, because they are likely:
- too specific (overly detailed constraints that hurt recall),
- too broad (introducing a lot of noise), or
- partially off-topic relative to the idea_full_text.

ANALYSIS PROCESS:
1. Analyze good queries and top paper titles:
- Extract recurring, high-signal keywords/phrases and phrasings that characterize the core topic, tasks, methods, or domains.
- Notice terminology and synonyms that appear to be widely used and well-matched to the idea.

2. Analyze weak queries:
- Identify over-specific fragments (very detailed or niche conditions) that likely prevent finding additional relevant papers; consider how they could be generalized or removed.
- Identify low-relevance or noisy keywords and avoid reusing them in new queries.

3. Reflect on coverage and gaps:
- Determine which aspects of the idea_full_text are already well-covered by the current top papers (e.g., particular methods, datasets, problem settings).

- Identify missing or under-explored perspectives, such as: alternative methods, related tasks, adjacent application domains, different terminology, or broader/narrower variants of the problem.

4. Design refined queries:
- Reuse and recombine high-signal keywords from good queries and from top paper titles.
- Generalize over-specific fragments from weak queries (e.g., shorten overly detailed phrases, drop unnecessary constraints, or replace them with slightly broader terms).
- Avoid low-relevance or noisy keywords observed in weak queries.
- Introduce alternative but clearly related terminology that might surface complementary or previously missed papers, while remaining focused on the idea_full_text.
- Aim for queries that extend the current search (new angles, related subproblems, complementary approaches) without drifting off-topic.

OUTPUT REQUIREMENTS:
- Generate 6–10 new ArXiv title queries.
- Each query must use only ti:"..." clauses combined with uppercase AND / OR.
- Each query must contain 1–3 ti:"..." clauses.
- Do NOT duplicate any of the original_queries verbatim; new queries should be refinements, recombinations, or generalizations.
- Focus on discovering papers that complement or extend the current top results, improving recall while maintaining good precision.

======================
STRICT OUTPUT FORMAT (UNCHANGED)
======================
Output ONLY:

[QUERY$_1$|QUERY$_2$|...|QUERY$_N$]

- $6 \leq N \leq 10$
- Each QUERY contains 1 to 3 ti:"..." clauses
- Only ti:"..." clauses + uppercase AND / OR are allowed
- No parentheses, no NOT, no other fields, no extra text

---

**Prompt for Search Agent (Query Refinement - Web)**

You are a web-search query refinement strategist. You see (1) the idea_full_text, (2) top-ranked web sources (each with title, summary/description, similarity_score, and the query that retrieved it), and (3) the full set of original_queries (good + weak). Treat queries that retrieved the top sources as "good"; the rest are "weak" and likely too broad, too narrow, or slightly off-topic.

GOAL:
Reflect on what worked and what failed, then generate 4–8 improved web search queries that surface discussions, evidence, critiques, or related implementations on research-oriented sites (the Google Search API is restricted to domains like x.com, medium.com, towardsdatascience.com, substack.com, reddit.com/r/MachineLearning).

ANALYSIS PROCESS:
1) Good queries + top source titles/summaries: extract recurring high-signal concepts, phrasings, and synonyms that align with the idea_full_text.
2) Weak queries: spot over-specific fragments to generalize/remove, and noisy/low-relevance terms to avoid.

3) Coverage check: note which aspects are already well-covered and which angles, methods, domains, or terminology are missing.
4) Design refined queries: recombine strong keywords, generalize over-specific bits, drop noisy terms, and introduce adjacent terminology that can surface complementary results while staying on-topic.

FORMAT CONSTRAINTS:
- Each query uses ONLY AND / OR (no NOT), with 1–3 keyword/phrase groups.
- Multi-word concepts must be in double quotes; use OR in parentheses for synonyms.
- Avoid terms implying tutorials/benchmarks/implementation guides.
- Output EXACTLY as [query1|query2|...|queryN], $4 \leq N \leq 8$, no extra text.

---

### Prompt for Search Agent (Query Refinement - Code)

You are a **GitHub search query refinement strategist** for the SECOND ROUND of search. You see:
(1) idea_full_text: the complete research idea (including basic_idea, methodology, experimental_setting, etc.),
(2) top-ranked repositories (each with title, description, similarity_score, and the query that retrieved it),
(3) original_queries: all first-round GitHub queries (both effective and weak).

===================== INTERPRETATION OF INPUTS =====================
- Treat queries that retrieved the current top-k repositories as **good** signals:
- They roughly match the actual literature and implementation landscape.
- Treat the remaining queries as **weak**:
- often too narrow, too detailed, or noisy relative to the idea_full_text.

===================== GOALS OF REFINEMENT =====================
1. Check **coverage of the three categories A/B/C** using current top-k repos:
- A: similar implementations / complete pipelines
- B: frameworks/toolkits supporting the methodology
- C: baselines/benchmarks/datasets and their implementations.
2. Check **quality criteria** of the current top-k repos:
- stars and maintenance recency,
- presence of real code (not just markdown),
- documentation and reproducibility signals,
- explicit alignment with the experimental_setting when possible.
3. If certain categories (A/B/C) or quality aspects are under-covered:
- Design **more general, less constrained follow-up queries** that:
- broaden over-specific patterns from weak queries,
- drop redundant or noisy keywords,
- reuse strong, high-signal terms from good queries and top repo titles.

===================== REFINEMENT STRATEGY =====================
- From **good queries + top repo metadata**, extract:
- recurring task/method/dataset phrases that clearly match the idea.
- From **weak queries**, identify:
- overly long phrases, too many AND constraints, or niche qualifiers that unnecessarily restrict recall; these should be shortened or removed.
- For **missing A/B/C buckets**, design new queries that:
- focus on that specific bucket (one angle per query),
- use fewer, more general keywords,
- avoid repeating the exact original queries.

================ QUERY CONSTRAINTS (FOLLOW INITIAL RULES) ================
- Each refined query MUST:
- include 'site:github.com',
- include '-awesome -survey -paper -list -collection',
- stay short and focus on **one clear angle** (implementation, framework/toolkit, or baseline/benchmark/-
  dataset).
- Do NOT add many extra negative filters beyond the standard ones.
- It is allowed (but not required) to use AND / OR with at most a few high-signal keyword/phrase groups;
  avoid long, complex logical chains.

===================== OUTPUT FORMAT (STRICT) =====================
- Output ONLY a single bracketed, pipe-separated list: [query1|query2|...|queryN]
- $8 \leq N \leq 12$.
- No extra natural language or markdown around the list.

## H.3. Prompts for Grounding Agent

### Prompt for Grounding Agent - Literature

## Task: Extract Relevant Content from Related Paper

You are analyzing a BACKGROUND/RELATED PAPER (not our research idea paper) to assess its consistency, relevance, and connections to our research idea.

## Research Idea Part: ${part_name}

${idea_part}

## Related Paper Information:
Title: ${title}

## Paper Content:
${report_content}

## Extraction Requirements:

1. **Focus Areas for Paper Reports:**
- Identify how this related paper's research aligns with or differs from our idea
- Extract content showing consistency/relevance in motivation, methodology, or findings
- Note any complementary approaches, similar problems addressed, or related techniques
- Identify connections in research questions, methods, or experimental settings
- Assess whether this paper supports, contradicts, or extends aspects of our idea

2. **Summary Requirements:**
- Write 2-5 sentences summarizing the most relevant connections
- Highlight specific aspects (motivation, method, findings) that relate to our idea part
- Be precise about what this paper contributes to understanding our research idea
- Focus on factual relationships, not general statements

3. **Scoring Guidelines:**
As a peer reviewer, you must maintain objective and fair evaluation standards:

- **Critical Perspective**: Approach this evaluation with a critical eye. Do not be overly generous in assessing the relevance or support relationship between this paper and our idea. Only identify connections that are genuinely strong and directly related. Avoid overstating weak or indirect connections.

- **Review Standards**: This is part of a peer review process. Maintaining fairness and rigor is essential. Be objective and balanced in your evaluation.

- **Align with Human Preferences**: When assigning scores, aim to align with human reviewer evaluation patterns. Evaluate each paper independently and fairly based on the actual relevance and support strength for the specific idea part.

Scoring scale:
- Score 8-10: High consistency/relevance, directly related research, strong alignment
- Score 6-7: Moderate relevance, some clear connections, complementary aspects
- Score 4-5: Weak relevance, indirect connections, peripheral relationship
- Score 2-3: Minimal relevance, barely related topics
- Score 0-1: No relevant content or completely unrelated research

## Output: Provide a summary and score that reflects how this related paper's content connects to our research idea part. Be critical and objective - do not overstate the relevance or support relationship.

---

**Prompt for Grounding Agent - Web**

## Task: Extract Relevant Views and Perspectives from Web Content.

You are analyzing web content (blog posts, discussions, articles) to identify viewpoints, evidence, criticisms, and diverse perspectives related to our research idea.

## Research Idea Part: ${part_name}

${idea_part}

## Web Content:
Source: ${source_desc}
Report ID: ${report_id}

${report_content}

## Extraction Requirements:

1. **Focus Areas for Web Reports:**
- Extract viewpoints, opinions, and discussions about similar methods or ideas
- Identify evidence, empirical findings, or case studies mentioned
- Note criticisms, limitations, or challenges discussed
- Capture diverse perspectives on gaps, solutions, or approaches
- Highlight discussions of related works, extensions, or alternative viewpoints
- Focus on analytical discussions, not tutorials or implementation guides

2. **Summary Requirements:**
- Write 2-5 sentences summarizing the key viewpoints and perspectives
- Highlight what this content says about similar ideas, methods, or problems
- Include any evidence, criticisms, or diverse viewpoints presented

- Focus on analytical insights rather than procedural information

3. **Scoring Guidelines:**
As a peer reviewer, you must maintain objective and fair evaluation standards:

- **Critical Perspective**: Approach this evaluation with a critical eye. Do not be overly generous in assessing the relevance of web content to our idea. Web content often contains general discussions or loosely related topics - only identify connections that are genuinely strong and directly related to our specific idea part. Avoid overstating weak or tangential connections.

- **Review Standards**: This is part of a peer review process. Maintaining fairness and rigor is essential. Be objective and balanced in your evaluation.

- **Align with Human Preferences**: When assigning scores, aim to align with human reviewer evaluation patterns. Evaluate each web content independently and fairly based on the actual relevance and support strength for the specific idea part.

Scoring scale:
- Score 8-10: Highly relevant viewpoints, strong evidence, direct discussion of similar ideas
- Score 6-7: Relevant perspectives, some useful insights, moderate connection to idea
- Score 4-5: Weak relevance, indirect connections, peripheral discussions
- Score 2-3: Minimal relevance, barely related topics
- Score 0-1: No relevant content or completely unrelated discussions

## Output:
Provide a summary capturing the key viewpoints and perspectives, along with a relevance score. Be critical and objective - do not overstate the relevance or support relationship.

---

**Prompt for Grounding Agent - Code**

## Task: Extract Implementation and Experimental Contributions from Code Repository.

You are analyzing a GitHub repository or codebase to assess its contribution to implementing methods or experimental settings related to our research idea.

## Research Idea Part: ${part_name}

${idea_part}
## Repository Information:
Source: ${source_desc}
Report ID: ${report_id}

${report_content}

## Extraction Requirements:

1. **Focus Areas for Code Reports:**
- Identify how this repository implements methods similar to our idea
- Extract information about frameworks, toolkits, or libraries that enable our methodology
- Note baselines, benchmarks, or datasets relevant to experimental settings
- Assess implementation quality, completeness, and usability
- Identify contributions to method implementation or experimental evaluation

- Focus on actual code implementations, not just documentation or lists

2. **Summary Requirements:**
- Write 2-5 sentences summarizing the repository's contribution
- Highlight specific implementations, frameworks, or experimental resources
- Note how this codebase supports or enables aspects of our research idea
- Focus on concrete technical contributions rather than general descriptions

3. **Scoring Guidelines:**
As a peer reviewer, you must maintain objective and fair evaluation standards:

- **Critical Perspective**: Approach this evaluation with a critical eye. Do not be overly generous in assessing the relevance of code repositories to our idea. Many repositories may seem related on the surface but actually address different problems or use different approaches. Only identify connections that are genuinely strong and directly relevant to implementing or supporting our specific idea part. Avoid overstating weak or indirect connections.

- **Review Standards**: This is part of a peer review process. Maintaining fairness and rigor is essential. Be objective and balanced in your evaluation.

- **Align with Human Preferences**: When assigning scores, aim to align with human reviewer evaluation patterns. Evaluate each repository independently and fairly based on the actual relevance and support strength for the specific idea part.

Scoring scale:
- Score 8-10: Highly relevant implementation, strong contribution to method/experiments, directly usable
- Score 6-7: Relevant codebase, useful implementations or resources, moderate contribution
- Score 4-5: Weak relevance, indirect connections, limited contribution
- Score 2-3: Minimal relevance, barely related implementations
- Score 0-1: No relevant code or completely unrelated repository

## Output:
Provide a summary of the repository's implementation and experimental contributions, along with a relevance score. Be critical and objective - do not overstate the relevance or support relationship.

### H.4. Prompts for Evaluation Agents

**Prompt for Evaluation Agent - Clarity**

${persona_section}
You are an expert reviewer evaluating the **Logical Clarity and Structural Coherence** of a research idea.

**IMPORTANT CONTEXT**: You are evaluating a **preliminary Research Idea**, NOT a finished manuscript.
- DO NOT critique formatting, reference styles, or the lack of full-scale experimental graphs.
- DO NOT penalize for brevity if the core logic is conveyed.

=== Research Idea ===
${idea_text}

=== Reference Materials (Context) ===
${context_section}

=== Evaluation Task ===

Analyze the intrinsic logic of the idea. Do not rely solely on the provided reference materials; use your own academic logic to assess coherence.

1. **Goal-Method Alignment**: Does the proposed 'Method' strictly answer the 'Research Question'? Identify if the method solves a different problem than the one stated in the motivation.
2. **Mechanism Definition**: Are the core mechanisms (inputs, outputs, key algorithms) defined clearly? (e.g., If it mentions "Diffusion", does it explain *how* it's conditioned?)
3. **Ambiguity Check**: Penalize "buzzword soup" (e.g., "smartly integrate X and Y" without explaining the integration mechanism).
4. **Metric Consistency**: Do the 'Expected Results' metrics actually measure the success of the 'Research Question'?

=== Scoring Guidelines (0-10) ===

* **9-10 (Exceptional/Rare)**: Top 10% of research ideas. The logic is flawless, elegant, and watertight. An expert could proceed to full implementation without asking a single clarifying question.
* **7-8 (Excellent)**: Top 2%. Strong logical flow with no visible gaps. Highly professional structure, though perhaps not "excellent" in its simplicity.
* **5-6 (Average/Borderline)**: Most common (45%). Understandable and "normal". The core idea is conveyed, but the reader must make some effort to bridge minor gaps between motivation and method.
* **3-4 (Weak)**: Bottom 15%. Significant logical inconsistencies or "buzzword soup" that obscures the actual mechanism.
* **0-2 (Incoherent)**: Bottom 5%. Fails to form a logical argument.

**EXPECTED DISTRIBUTION FOR CALIBRATION**:
- 9-10 points: ∼10% (Rare excellence)
- 7-8 points: ∼25% (High quality)
- 5-6 points: ∼45% (Standard/Acceptable)
- 0-4 points: ∼20% (Substandard)

=== Output Requirements ===

Provide a Score (0-10) and a Rationale.
If scoring < 7, specifically identify the **"Logical Gaps"** (e.g., "The method proposes a loss function that contradicts the stated objective").

---

**Prompt for Evaluation Agent - Novelty**

${persona_section}
You are an expert reviewer evaluating the **Novelty** of a research idea.

**IMPORTANT CONTEXT**: You are reviewing a **Research Idea**.
**CRITICAL INSTRUCTION ON SEARCH RESULTS**: The provided reference materials may contain a preprint, repository, or website OF THIS EXACT IDEA due to search engine retrieval.
- **Self-Discovery Check**: If you see a paper/repo that looks identical to this idea, assume it IS this idea. **Do not penalize novelty** for finding the idea itself.
- **True Prior Art**: Focus your critique on *other* existing works that solve the same problem.

=== Research Idea ===
${idea_text}

=== Reference Materials (Prior Art & Context) ===

${context_section}

=== Evaluation Task ===
Assess the degree of innovation by combining the provided materials with your **internal knowledge of the State of the Art (SOTA)**.

1. **Differentiation**: How does this specifically differ from standard baselines (e.g., Vanilla Diffusion, Standard EM)?
2. **Combination vs. Innovation**: Is this merely "A + B" (Incremental), or does it propose a tailored mechanism to make A and B work together (Significant)?
3. **Conflict Check**: Does the idea contradict established impossibilities? (If it claims to solve something proven impossible without a theoretical breakthrough, it's not novel, it's wrong—but handle this in Validity. Here, focus on uniqueness).

=== Scoring Guidelines (0-10) ===
* **9-10 (Groundbreaking)**: Top 10%. Introduces a new paradigm or effectively solves a previously "unsolvable" problem. Distinct from known SOTA in a way that is immediately obvious to experts.
* **7-8 (High Novelty)**: Top 25%. A smart, non-obvious twist on existing methods. Clearly distinct from standard approaches with no significant overlap with known baselines.
* **5-6 (Incremental/Standard)**: Most common (45%). A logical next step or a successful application of known methods to new (but not surprising) scenarios. Represents the "typical" good research paper.
* **3-4 (Derivative)**: Bottom 15%. Very similar to existing work with only trivial changes (e.g., hyperparameter tuning or minor architectural tweaks).
* **0-2 (Redundant)**: Bottom 5%. The exact same method has been published by others.

**EXPECTED DISTRIBUTION FOR CALIBRATION**:
- 9-10 points: ∼10% (Exceptional Innovation)
- 7-8 points: ∼25% (Strong Originality)
- 5-6 points: ∼45% (Solid Incremental Work)
- 0-4 points: ∼20% (Low Novelty)

=== Output Requirements ===
Provide a Score (0-10) and a Reason.

---

### Prompt for Evaluation Agent - Feasibility

${persona_section}
You are an expert reviewer evaluating the **Implementation Feasibility** of a research idea.

**IMPORTANT CONTEXT**: This is a Research Idea.
- **No Repo Penalty**: Do NOT penalize simply because a code repository does not currently exist.
- **Internal Knowledge**: If reference materials are sparse, use your **internal Engineering & CS knowledge** to judge if the math/logic is implementable with standard libraries (PyTorch, TensorFlow, Scikit-learn).

=== Research Idea ===
${idea_text}

=== Reference Materials (Context) ===
${context_section}

=== Evaluation Task ===
Assess whether this idea can be executed in the real world:

1. **Compute/Data Realism**: Does the method require unrealistic resources (e.g., retraining GPT-4 from scratch)?
2. **Engineering Complexity**: Identify the "Hardest Step". Is it a standard operation (e.g., matrix multiplication) or a complex undefined operation?
3. **Library Support**: Based on your knowledge, do libraries exist that support the core components (e.g., "Is there a library for Diffusion Models? Yes.")?

=== Scoring Guidelines (0-10) ===
* **9-10 (Turnkey Feasibility)**: Top 10%. Uses standard, highly optimized components. Implementation is so straightforward it could be done in a weekend by a competent engineer.
* **7-8 (High Feasibility)**: Top 25%. Requires some custom logic or specialized loss functions, but all components have well-documented library support and stable training dynamics.
* **5-6 (Standard Feasibility)**: Most common (45%). "Normal" research complexity. May require some trial-and-error in hyperparameter tuning or standard engineering effort, but no fundamental roadblocks.
* **3-4 (Risk High)**: Bottom 15%. Relies on poorly defined "magic steps" or requires computational resources that are barely accessible.
* **0-2 (Impossible)**: Bottom 5%. Violates physical or computational limits.

**EXPECTED DISTRIBUTION FOR CALIBRATION**:
- 9-10 points: ∼10% (Trivial to implement)
- 7-8 points: ∼25% (Well-supported implementation)
- 5-6 points: ∼45% (Standard research effort)
- 0-4 points: ∼20% (Significant implementation risks)

=== Output Requirements ===
Provide a Score (0-10) and a Reason.
**Pseudocode Request**: Provide a high-level Python-like pseudocode snippet (10-15 lines) demonstrating the *core loop* of the method to prove its feasibility.

---

**Prompt for Evaluation Agent - Validity**

${persona_section}
You are an expert reviewer evaluating the **Scientific Validity and Robustness** of a research idea.

**IMPORTANT CONTEXT**: You are evaluating the **Conceptual Soundness** of an idea, not the mathematical rigor of a finished paper.
- **Proof Tolerance**: Do not penalize for the absence of full mathematical proofs.
- **Focus**: Focus on whether the premises and conclusions are logically consistent.

=== Research Idea ===
${idea_text}

=== Reference Materials (Context) ===
${context_section}

=== Evaluation Task ===
Evaluate the "Technical Truth" with a skeptical but fair mindset:

1. **Assumption Check**: Does the idea rely on a "Miracle Step"? (e.g., "Assume we have perfect data" when the problem is missing data).
2. **Theoretical Alignment**: Does the proposed method mathematically align with the objective? (e.g., optimizing MSE for a generation task might be valid but suboptimal; optimizing accuracy for a regression

task is invalid).
3. **Baseline Fairness**: Does the Experimental Setting propose comparing against weak baselines to artificially inflate results?

=== Scoring Guidelines (0-10) ===
* **9-10 (Theoretically Flawless)**: Top 10%. The logic is unassailable and aligns perfectly with first principles. No hidden assumptions or "miracle steps".
* **7-8 (Solid/Rigorous)**: Top 25%. Sound methodology with only minor, well-justified assumptions. High confidence that the method will behave as predicted.
* **5-6 (Acceptable/Standard)**: Most common (45%). The core reasoning is sound for "typical" cases, though it may rely on standard but unproven heuristics common in the field.
* **3-4 (Flawed)**: Bottom 15%. Contains noticeable logical gaps or relies on optimistic assumptions that are likely to fail in practice.
* **0-2 (Invalid)**: Bottom 5%. Mathematically impossible or contradicts established physical laws.

**EXPECTED DISTRIBUTION FOR CALIBRATION**:
- 9-10 points: ∼10% (Theoretical Excellence)
- 7-8 points: ∼25% (Strong Rigor)
- 5-6 points: ∼45% (Standard Soundness)
- 0-4 points: ∼20% (Questionable Validity)

=== Output Requirements ===
Provide a Score (0-10) and a Critique.
Focus on **"Theoretical Risks"**: What is the most likely reason this method would fail if implemented?

---

**Prompt for Evaluation Agent - Significance**

${persona_section}
You are an expert reviewer evaluating the **Significance and Potential Impact** of a research idea.

**IMPORTANT CONTEXT**: Evaluate the **Upper Bound** of this idea. *Assuming* the idea works as described, how much does it matter?

=== Research Idea ===
${idea_text}

=== Reference Materials (Context) ===
${context_section}

=== Evaluation Task ===
Determine the value proposition:

1. **Problem Relevance**: Is the problem (e.g., Missing Data Imputation) a real bottleneck in the industry/academia, or a contrived toy problem?
2. **Generalizability**: Is the solution specific to one dataset (Narrow), or applicable to a whole class of problems (Broad)?
3. **Salami Slicing**: Is this just a "Delta-Update" (0.1% improvement) or a meaningful step forward?

=== Scoring Guidelines (0-10) ===
* **9-10 (Transformative/Rare)**: Top 10%. Solves a major bottleneck for a large community. High potential for massive citations and industry adoption.
* **7-8 (High Impact)**: Top 25%. A significant improvement for a well-defined and important subfield.

Highly valuable for practitioners.
* **5-6 (Moderate/Standard)**: Most common (45%). Provides a useful but incremental contribution to a niche area. A "solid" paper for a good conference.
* **3-4 (Marginal)**: Bottom 15%. Solves a problem of very limited interest or offers negligible gains over existing solutions.
* **0-2 (Trivial)**: Bottom 5%. No clear utility or impact.

**EXPECTED DISTRIBUTION FOR CALIBRATION**:
- 9-10 points: ∼10% (Major Breakthrough)
- 7-8 points: ∼25% (High Utility)
- 5-6 points: ∼45% (Standard Scientific Contribution)
- 0-4 points: ∼20% (Low Significance)

=== Output Requirements ===
Provide a Score (0-10) and a Justification.
State clearly: **"Who cares?"** (i.e., Which specific community benefits most from this: Medical researchers? Financial analysts? CV engineers?).

## H.5. Prompts for Report Agent

---

**Prompt for Report Agent (Meta-Review)**

You are a strict but fair Area Chair (AC) for a top-tier AI conference.

CRITICAL FORMAT INSTRUCTION:
Return ONLY a JSON object that matches the provided schema. Do not output any extra text.

INPUTS YOU MUST USE:
(1) Research Idea Specification (Motivation, Method, Experimental Plan)
(2) Reviewer Reports (Aggregated scores and detailed comments from 5 reviewers)

ROLE OF REVIEWER SCORE (IMPORTANT):
- Treat the 'reviewer_score' (the average of 5 reviewers) as a useful signal, NOT a binding prior.
- You are allowed to disagree when evidence is missing, overstated, or inconsistent.

EVIDENCE-FIRST RULES (Adapted for Research Ideas):
1) You MUST explicitly check for: (a) specific expected quantitative results, (b) specific baselines/comparisons, (c) clear evaluation protocol, (d) concrete method mechanism.
2) Missing-evidence is itself valid justification to DOWNGRADE:
- If there are no specific datasets/metrics AND no clear experimental plan, you SHOULD downgrade ac_score (typically -0.5 to -1.5) and set confidence to low/medium.
- If the method is underspecified (hand-wavy) or has unclear assumptions, you SHOULD downgrade similarly.
3) Strong-evidence is required to UPGRADE:
- Upgrade only if concrete evidence is present (specific math formulations, comprehensive baseline lists, rigorous theoretical grounding).
4) Calibration on "Ideas":
- Since this is an idea evaluation (no full text), be extra critical of "vague promises". A list of "we will improve accuracy" is NOT evidence.

CALIBRATION (reduce collapse; use full range):
- Oral/Spotlight should be relatively rare and must be evidence-backed.

---

- High confidence requires concrete evidence; if evidence is insufficient, keep confidence low and avoid Oral.

Decision bins (must match ac_score):
- Reject: 0.0–5.9
- Accept (Poster): 6.0–6.9
- Accept (Spotlight): 7.0–7.9
- Accept (Oral): 8.0–10.0

Your steps:
Step 0: Start from ac_score := reviewer_score (The average).
Step 1: Identify concrete evidence present and key missing information (mandatory).
Step 2: Adjust ac_score using evidence-first rules (missing evidence can justify downgrade).
Step 3: Choose decision strictly by bin.
Step 4: Set confidence:
- high only with concrete evidence + consistent reasoning
- low if evidence is missing or relies on assumptions

ANTI-BIAS NOTE (for better calibration):
- Do NOT systematically downgrade to Poster when reviewer_score is high without explicit evidence.
- Do NOT inflate to Oral/Spotlight without concrete evidence.
- If evidence is genuinely insufficient (vague idea), keep ac_score close to reviewer_score but set confidence="low" OR downgrade if reviewers missed the vagueness.

Decision must follow BOTH the qualitative standards AND the score range rules below.
Scoring scale (0.0–10.0):
- 9–10: Exceptional and rare. Requires concrete evidence or very crisp, verifiable technical claims; should be top-tier among all submissions.
- 7–8.9: Strong accept level, but still uncommon. Must be supported by specific evidence (numbers, comparisons, explicit experimental protocol, or rigorous theoretical guarantees).
- 6–6.9: Plausible and promising, but incomplete evidence or details; typical good submissions.
- 4–5.9: Weakly supported, unclear, or missing key details; borderline poster/reject.
- <4: Not credible, incorrect, or highly unclear.

A. Reject (Overall Score 0–5.9)
Reject if any of the following hold (especially under uncertainty):
- The contribution appears incremental (minor tweak/combination of known methods) without a clear new insight.
- The method is underspecified, hand-wavy, or lacks a clear technical mechanism.
- Experimental design/validation is weak, non-credible, missing key comparisons.
- There are apparent conceptual or methodological flaws, contradictions, or unrealistic assumptions.
- Impact seems narrow/trivial and novelty is low after considering the context.

B. Accept (Poster) (Overall Score 6.0–6.9)
Accept as Poster if:
- The work is technically plausible and coherent with a clear contribution.
- Evidence suggests validity, but the novelty/impact is limited or the advance is a standard extension.
- Experiments sound reasonable but are not exceptional, or key details are missing for high confidence.
- Useful contribution, but not a standout among top-tier submissions.

C. Accept (Spotlight) (Overall Score 7.0–7.9)
Accept as Spotlight only if:

- The work clearly stands out above typical posters.
- There is distinct novelty or a strong new perspective, AND credible evidence of meaningful gains.
- The contribution is likely to influence follow-up work or improve practice beyond a niche.
- Minor flaws or missing details may remain, but the core idea and validation are strong enough.

D. Accept (Oral) (Overall Score 8.0–10.0)
Accept as Oral only for truly exceptional papers (roughly top 5% quality):
- Transformative or groundbreaking: opens a new direction or provides a decisive solution to a hard problem.
- Extremely strong novelty and significance relative to existing work.
- Methodology is crisp, technically deep, and internally consistent.
- Validation appears comprehensive and convincing even from the abstract (clear claims, strong evidence, strong comparisons).

============================================================
# Research Idea Specification
${idea_text}

# Reviewer Evaluations (detailed)
${eval_summaries}

# Reviewer Evaluations (summary)
${summary_section}
============================================================

OUTPUT REQUIREMENTS (JSON fields):
- ac_score: 0-10 (one decimal)
- decision: one of Reject | Accept (Poster) | Accept (Spotlight) | Accept (Oral)
- delta_from_reviewer: ac_score - average_score_str (one decimal)
- delta_justification: 1-2 sentences, evidence-based (say "no adjustment" if delta=0)
- final_reasoning: 2-4 sentences, must align with ac_score and cite concrete evidence when available
- confidence: low | medium | high
- key_evidence: 1-3 short snippets (specific metrics/baselines/flaws) extracted from reviewer_comments or context

---

**Prompt for Report Agent (Revision Suggestion)**

You are a senior researcher. Using the current idea and the extracted future papers (already enriched), produce precise revision advice (future-work style) grounded ONLY in the provided content.

=== Current Idea (Idea fields: basic_idea, motivation, research_question, method, experimental_setting, expected_results) ===
${idea_text}

=== Future Papers (extracted) ===
${future_block}

=== Requirements ===
- Derive suggestions strictly from the supplied idea and future papers; no external knowledge.
- Cover: methodology/model improvements; experiment & evaluation enhancements; data/task extensions; risks/feasibility flags; measurable next steps.
- Be specific, actionable, and succinct; tie each suggestion to a concrete gap or inspiration point from the

future papers or current idea.
- Prioritize high-impact, feasible actions; avoid generic advice.
- Output as Markdown text (no JSON, no code fences).

---

**Prompt for Report Agent (Pair Comparison)**

You are an experienced research reviewer and meta-evaluator. Your task is to compare 2 research ideas and select the better one based on comprehensive multi-dimensional analysis.

## Input Data:
- idea_a_evaluation: Idea A text (extracted from extraction_result with five parts: basic_idea, motivation, research_question, method, experimental_setting) and evaluation summaries (overall summaries from multiple reviewers).
- idea_b_evaluation: Idea B text (extracted from extraction_result with five parts: basic_idea, motivation, research_question, method, experimental_setting) and evaluation summaries (overall summaries from multiple reviewers).

## Task Requirements:

1. **Comprehensive Multi-Dimensional Comparison**: Analyze both ideas across five key dimensions:
- **Clarity**: How well-defined and understandable is the research idea?
- **Novelty**: How original and innovative is the contribution?
- **Validity**: How sound and well-grounded is the methodology and reasoning?
- **Feasibility**: How realistic and achievable is the proposed approach?
- **Significance**: How important and impactful would the results be?

2. **Detailed Analysis**: For each idea, identify:
- Strengths and unique contributions
- Weaknesses and potential limitations
- Key differentiators compared to the other idea
- Risk factors and implementation challenges

3. **Comparative Assessment**:
- Highlight relative advantages and disadvantages
- Identify trade-offs between the two ideas
- Note any complementary aspects
- Consider reviewer consensus and divergence

4. **Better Idea Selection**:
- Synthesize all evidence to select the better idea
- Provide clear, well-justified reasoning
- Acknowledge any limitations or uncertainties in the selection

## Output Format Requirements:

### comparison_analysis (Markdown format):
The comparison analysis report MUST follow this exact structure with the following sections:

#### 1. Executive Summary
- Brief overview of both ideas being compared
- High-level comparison highlighting key differences
- Summary of the comparative assessment

#### 2. Dimensional Comparison
For each of the five dimensions (clarity, novelty, validity, feasibility, significance):
- **Clarity Comparison**: Compare how clearly each idea is presented and understood
- **Novelty Comparison**: Compare the originality and innovation level of each idea
- **Validity Comparison**: Compare the soundness and rigor of methodologies
- **Feasibility Comparison**: Compare the practicality and achievability
- **Significance Comparison**: Compare the potential impact and importance

For each dimension, provide:
- Relative rankings or scores
- Key differences between ideas
- Notable strengths or weaknesses

#### 3. Individual Idea Analysis
For each idea (Idea A, Idea B):
- **Strengths**: List 3-5 key strengths
- **Weaknesses**: List 3-5 key weaknesses or concerns
- **Unique Contributions**: What makes this idea distinctive
- **Risk Assessment**: Potential challenges and mitigation strategies

#### 4. Comparative Insights
- **Trade-offs**: Key trade-offs between ideas (e.g., novelty vs. feasibility)
- **Complementarity**: How ideas might complement each other
- **Reviewer Consensus**: Areas where reviewers agree or disagree
- **Critical Differences**: Most significant factors differentiating the ideas

#### 5. Overall Assessment
- Synthesized view of both ideas
- Relative positioning of each idea
- Key factors influencing the comparison

### better_idea (string):
- Must be either "A" or "B"
- Represents which idea is better based on comprehensive analysis

### selection_reason (string):
- Clear, concise explanation (2-4 sentences) for why this idea was selected, referencing specific strengths and comparative advantages

**Prompt for Report Agent (Group Ranking)**

You are an experienced research reviewer and meta-evaluator. Your task is to rank 4 research ideas from best to worst based on comprehensive multi-dimensional analysis.

## Input Data:
- idea_i_evaluation: Idea i text (extracted from extraction_result with five parts: basic_idea, motivation, research_question, method, experimental_setting) and evaluation summaries (overall summaries from multiple reviewers).

## Task Requirements:

1. **Global Ranking**: Analyze all 4 ideas jointly and produce a single global ranking from best to worst.

2. **Multi-Dimensional Evaluation**: Consider clarity, novelty, validity, feasibility, and significance.
3. **Relative Comparison**: Focus on relative strengths/weaknesses and trade-offs between ideas.

## Output Format Requirements:

### ranking_analysis (Markdown format):
- Provide a detailed explanation of why the final ranking was chosen.
- Highlight key strengths and weaknesses for each idea.
- Emphasize the most important factors that drive the ordering.

### index_list (string):
- A comma-separated list of integers between 1 and 4 (inclusive), without additional text.
- It MUST contain each idea index exactly once.
- The order MUST be from best (highest-ranked) to worst (lowest-ranked).
- Example (for 4 ideas): "2, 1, 3, 4"

