# OpenReview forum: "InnoEval: On Research Idea Evaluation as a Knowledge-Grounded, Multi-Perspective Reasoning Problem"
_ICML.cc/2026/Conference — ICML 2026 regular_

### Official Review · Reviewer_8UoA · 2026-03-07

**Soundness:** 2
**Presentation:** 3
**Significance:** 2
**Originality:** 2
**Overall Recommendation:** 4
**Confidence:** 4

**Summary:**

The paper proposes InnoEval, a pipeline that evaluates research ideas by retrieving and grounding evidence from papers, the web, and code, then aggregating multi-criteria scores from a committee of reviewer personas into a final meta-review. It also introduces OpenReview-based benchmarks and reports improvements over RAG and agent baselines on classification and ranking.

**Compliance With Llm Reviewing Policy:**

Affirmed.

**Final Justification:**

The paper proposes InnoEval, a pipeline that evaluates research ideas by retrieving and grounding evidence from papers, the web, and code, then aggregating multi-criteria scores from a committee of reviewer personas into a final meta-review. It also introduces OpenReview-based benchmarks and reports improvements over RAG and agent baselines on classification and ranking.
I proposed several concerns during the first round review, the authors' response addressed all my concerns. I will keep my score with a weak accept.

**Key Questions For Authors:**

1.How do you justify using the final decision label as a proxy for idea quality or novelty, rather than writing quality, experimental completeness, or track effects? Can you run controlled tests using only a minimal “core idea” description?
2.Please report systematic curves over the number of personas, retrieval rounds N, and retained items m, including wall-clock latency and monetary cost, and provide a recommended default configuration.
3.You note cases where a single persona underperforms a no-persona baseline for ranking; why does this happen, and can you propose a more controlled persona selection strategy than random sampling?

**Limitations:**

The pipeline is resource-intensive, requiring multiple stages of retrieval, slow content processing, grounding, and multi-persona scoring, which can translate into substantial API usage and monetary/compute overhead and may hinder practical deployment at scale.

**Strengths And Weaknesses:**

strengths：
1.The paper is well-motivated and clearly identifies key gaps in research idea evaluation, including limited grounding, weak deliberation, and insufficient multi-criteria assessment.
2.It introduces OpenReview-based benchmarks spanning pointwise decisions, pairwise comparisons, and groupwise selection/ranking, enabling systematic evaluation beyond single-score judging.
3. Ablations show that grounding, reviewer personalization, and web/code sources each contribute materially to performance.

weaknesses：
1.The method is primarily a pipeline of heuristic components with fixed hyperparameters, and lacks a principled objective or theoretical analysis explaining when the hybrid scoring and multi-agent aggregation are reliable.
2.Although the system uses heterogeneous online sources and LLM-based quality signals (citations, venue, popularity, stars) for ranking/filtering, these proxies do not guarantee factual correctness or robustness to noisy/manipulated web content, and the paper provides limited analysis of failure modes.
3.Many quality claims rely on LLM as a judge style evaluations, which risks an evaluation feedback loop and weakens measurement credibility.

---

> ### Author Rebuttal · Authors · 2026-03-30
>
> Dear Reviewer 8UoA,
>
> We are deeply grateful for your valuable time and insightful feedback. Below are our detailed responses to your concerns:
>
> # W1. Ablations on hybrid scoring and multi-agent aggregation
>
> Here, we present the ablation study on hybrid scoring based on the point-wise evaluation task:
>
> | |F1_2|F1_3|
> |---|---|---|
> |$\alpha=1$|60.83|62.21|
> |$\alpha=0.8$|67.74|66.36|
> |$\alpha=0.6$|71.43|70.51|
> |$\alpha=0.4$|72.35|71.89|
> |**$\alpha=0.2$ (InnoEval)**|**75.74**|**74.56**|
> |$\alpha=0$|74.19|72.81|
>
> The results demonstrate that LLM scoring outperforms embedding-based scoring, indicating that relying solely on semantic similarity tends to favor surface-level lexical patterns rather than substantive relevance, quality, and reputation. However, relying entirely on LLM ($\alpha=0$) incurs a slight degradation compared to the optimal hybrid configuration, confirming that inherent LLM biases and hallucinations can negatively impact evaluation quality.
>
> Regarding ablation studies on individual agents within InnoEval, please refer to our response to **Reviewer GWZN (W1&Q1: Most Responsible Components for the Gains)**, where we provide detailed experimental results and analysis.
>
> # W2. Noisy/manipulated web content
>
> To control web retrieval quality, we implement three measures: (1) restrict web sources to high-quality blogs and forums sites (e.g., Medium, Reddit machine learning); (2) filter noise and only retain relevant web content during enriched knowledge report generation via LLM-based distillation; (3) perform fine-grained grounding that aligns only core evidence with specific idea parts, accompanied by explicit relevance analysis.
>
> # W3. Many quality claims rely on LLM as a judge
>
> To mitigate LLM-as-judge biases, we employ GPT-4o-mini for qualitative evaluation, distinct from the DeepSeek-V3.2 backbone used in main experiments. Additionally, we conduct comprehensive **human evaluation** (Fig.2, Heat Map) from two perspectives: direct expert scoring and peer-review-based assessment. Our multi-faceted validation: quantitative, qualitative, and human ensures no single evaluation paradigm dominates our conclusions. Furthermore, Figure 3(c) demonstrates that InnoEval's evaluation reports effectively improve idea quality under third-party benchmarks, corroborating their practical utility beyond internal metrics.
>
> # Q1. Predicting labels vs. real innovation evaluation
>
> Please first refer to our response to **Reviewer KQBH (W1: Predicting labels vs. real innovation evaluation)** for detailed elaboration.
>
> We wish to underscore that **idea evaluation is holistic, not solely novelty**. A seemingly "groundbreaking" idea may sound highly novel, yet if it lacks feasibility, it cannot constitute a valuable research contribution. This multi-dimensional philosophy is precisely what distinguishes InnoEval from simplistic novelty-focused evaluation methods.
>
> We also conducted controlled tests using only **motivation** and **method** as the "**core idea**". Additionally, we sampled 50 point-wise core ideas for human evaluation **specifically on the Novelty** dimension and measured Pearson coherence with all methods:
>
> | |Pair| |Human Novelty Eval|
> |---|---|---|---|
> | |Acc easy|Acc hard|Coherence|
> |CoT|45.35|38.00|0.29|
> |RAG|47.67|40.50|0.35|
> |GraphEval|43.02|42.50|0.33|
> |ResearchAgent|54.65|46.00|0.42|
> |InternAgent|65.12|57.50|0.53|
> |ScholarEval|68.02|63.50|0.58|
> |**InnoEval**|**76.74**|**69.50**|**0.72**|
>
> Results demonstrate InnoEval can identify genuine novelty even with minimal core descriptions.
>
> # Q2. Systematic curves over persona nums, N and m
>
> We provide curves for persona numbers, N, and m regarding performance, wall-clock latency, and monetary cost **at this link: [URL](https://anonymous.4open.science/r/InnoEval-AC7E/rebuttal.pdf)**. Based on our experimental results, we recommend the default configuration: **Persona Numbers = 5, N = 3, m = 10**.
>
> # Q3. More controlled persona selection strategy
>
> The shift from 0 to 1 persona merely transforms the inherent bias of the LLM to that specific persona. It is difficult to disentangle whether model bias or persona bias causes more harm, which explains why single-persona evaluation underperforms the no-persona baseline in certain cases.
>
> Post-submission, we devised an improved persona selection strategy: **we input all personas to the LLM and prompt it to select the most suitable subset for each specific idea**. Experimental results are as follows:
>
> | | Point | |
> |---|---|---|
> | |F1 2|F1 3|
> |LLM select|78.19|78.94|
> |InnoEval|75.74|74.56|
>
> This strategy proves effective and will be incorporated into the updated InnoEval pipeline for further optimization.
>
>
>
> Thank you again for your constructive suggestions!
>
> **Please let us know if you have any further questions. If you find that our response addresses some of your concerns, would you kindly consider raising your rating score for our paper? We greatly appreciate your consideration.**
>
> Best Regards,
>
> Authors of InnoEval

---

> > ### Author Rebuttal · Reviewer_8UoA · 2026-04-03
> >
> > I have no further questions and will maintain my score with a weak accept.

---

> > > ### Author Response · Authors · 2026-04-03
> > >
> > > Dear Reviewer 8UoA,
> > >
> > > We once again express our sincere gratitude for your valuable time and effort. Your review comments have been immensely helpful in improving the quality of our paper.
> > >
> > > **We would like to cordially invite you to consider raising your rating score for our paper, given that you mentioned our rebuttal has fully addressed your concerns**. We have supplemented extensive experiments, including ablation studies on $\alpha$, persona num, N, and m, evaluations targeting the core idea, and explorations of better persona selection strategies. We believe our efforts are worthwhile. Please feel free to remind us if you have any further concerns, as your feedback is very important to us.
> > >
> > > Thank you once again for your dedication! We wish you all the best in your research and life.
> > >
> > > Best Regards,
> > >
> > > Authors of InnoEval

---

### Official Review · Reviewer_U5FS · 2026-03-08

**Soundness:** 2
**Presentation:** 2
**Significance:** 2
**Originality:** 3
**Overall Recommendation:** 4
**Confidence:** 4

**Summary:**

This paper introduces InnoEval, a framework for evaluating research ideas as a knowledge-grounded, multi-perspective reasoning problem. The method first extracts a structured representation of an idea, then performs heterogeneous online retrieval over papers, web content, and code repositories, followed by grounding of retrieved evidence to idea components. Evaluation is then carried out by a multi-agent “innovation review board” with different personas, across five dimensions: clarity, novelty, validity, feasibility, and significance. The framework outputs point-wise decisions, pair-wise comparisons, and group-wise rankings. To benchmark the approach, the authors construct datasets from ICLR 2025 and NeurIPS 2025 submissions by extracting idea tuples from papers and using conference decisions as labels. Empirically, the paper reports large gains over several baselines on point-wise, pair-wise, and group-wise tasks, and also presents human-correlation, ablation, and analysis results.

**Compliance With Llm Reviewing Policy:**

Affirmed.

**Key Questions For Authors:**

**Q1.** Could you provide a comparison of token usage, API cost, and latency across all baselines? How can one disentangle whether the reported gains stem from a better framework design or simply from using substantially more compute?

**Q2.** Has the extraction accuracy of the Extraction Agent been independently measured? How frequently was manual correction needed, and what types of errors were most common?

**Q3.** Conference decisions reflect many factors beyond idea quality, such as writing quality and experimental completeness. How would you disentangle whether InnoEval is evaluating the intrinsic quality of ideas versus predicting signals correlated with paper acceptance?

**Q4.** Could you share the prompt and win/lose judgment criteria used for the LLM-as-judge comparison in Table 2?

**Q5.** Is there a specific protocol to prevent the system from retrieving the source paper itself or its OpenReview page (including reviews and decisions) during online search? Do you consider timestamp filtering alone sufficient to prevent such leakage?

**Limitations:**

“Yes”

**Strengths And Weaknesses:**

**Strengths**

The paper studies an important and timely problem: how to evaluate research ideas in a more grounded and systematic way. The overall pipeline is ambitious and reasonably well designed, combining structured extraction, retrieval, grounding, and multi-perspective judging into a unified framework. The experiments are also fairly broad, covering point-wise, pair-wise, and group-wise evaluation settings.


**Weaknesses**

**W1. Efficiency and practicality are underdeveloped.**

The paper states that evaluating one sample costs $0.42, while the limitations section mentions that evaluating a single sample takes about half an hour. At the same time, the paper also suggests high throughput through parallelization. These claims are not clearly reconciled. I would like to see a more explicit comparison against baselines in terms of token usage, API cost, latency, and compute assumptions. As written, it is difficult to determine whether the reported gains come from a genuinely better evaluation framework or simply from using substantially more compute and tokens. In its current form, the practicality and deployment cost of the method are not convincingly established.

**W2. The Extraction Agent is never independently evaluated despite being a critical upstream dependency.**

The entire pipeline relies on this module to accurately distill papers into structured idea tuples, yet the paper offers no assessment of extraction quality. No extraction accuracy, inter-annotator agreement, or correction criteria are reported. Since errors here propagate through all downstream modules, this is a notable gap.

**W3. The benchmark target is confounded.**

The labels are derived from final conference decisions such as Reject / Poster / Spotlight / Oral. However, conference outcomes reflect much more than the intrinsic quality of an idea: they are also influenced by writing quality, experimental strength, implementation quality, reviewer variance, topic fit, and many other external factors. As a result, even if the extraction module successfully isolates an “idea tuple,” the supervision target remains only a noisy proxy for idea quality. This makes me uncertain whether the method is actually learning to evaluate research ideas, or instead learning to predict signals correlated with paper acceptance.

**W4. The qualitative evaluation lacks transparency and reproducibility.**

Table 2 uses o4-mini as an LLM judge to compare report quality across five criteria (Rationality, Supportiveness, Depth, Constructiveness, Overall Quality), but the paper does not disclose the judge prompt, evaluation rubric, or win/lose decision criteria. Appendix H includes prompts for all other agents but omits the one used for this qualitative comparison. This is particularly ironic given that the paper itself identifies LLM-as-a-Judge bias as a core problem in the introduction. Without these details, the reported win rates are neither reproducible nor verifiable.

**W5. There is a serious leakage / shortcut risk.**

This is, in my view, the paper’s most critical weakness. The method explicitly relies on online retrieval over literature, web content, and code repositories, and the paper emphasizes the benefit of using “living knowledge.” However, many benchmark instances are derived from papers that may already exist online, together with public metadata, repositories, discussions, or even OpenReview pages. I did not find a convincing protocol that prevents the system from retrieving the source paper itself or other public information that is strongly correlated with acceptance outcomes. If such leakage exists, then the reported improvements may be substantially overstated, because the model could be exploiting shortcut signals rather than genuinely evaluating the underlying idea.

---

> ### Author Rebuttal · Authors · 2026-03-30
>
> Dear Reviewer U5FS,
>
> We are deeply grateful for your valuable time and insightful feedback. Below are our detailed responses to your concerns:
>
> # W1&Q1. Efficiency and practicality
>
> Beyond data-level parallelization, InnoEval's pipeline can be further optimized by extensive intra-task parallelism: 1) concurrent search/filtering/iteration across paper/web/code sources, 2) parallel grounding of resources to idea components, and 3) simultaneous multi-persona/multi-metric evaluation. With maximal parallelization, per-sample costs compared to baselines are:
>
> | |token usage|API cost|time|
> | ------------- | ----------- | -------- | ----- |
> |CoT|0.09M|0.03$|102s|
> |RAG|0.28M|0.09$|254s|
> |ResearchAgent|0.46M|0.14$|667s|
> |InternAgent|0.42M|0.12$|395s|
> |ScholarEval|1.25M|0.37$|1248s|
> |InnoEval|1.41M|0.42$|1052s|
>
> Additionally, we would like to emphasize that a superior framework and increased computational budget are not mutually exclusive. **The additional tokens (vs. ScholarEval) primarily fund multi-perspective evaluation**, directly addressing the critical limitation of single-judge LLMs: inherent bias and limited knowledge horizons. As shown in Figure 3(a), **with comparable token budgets, our approach significantly outperforms naive test-time scaling**, demonstrating that gains stem from superior architectural design rather than mere computation increase.
>
> # W2&Q2. Extraction Agent Evaluation
>
> We have systematically evaluated the Extraction Agent on our 217 point-wise samples. Our quality criteria require that extracted content must be **self-contained, with motivation and methodology precisely aligned to the source paper; experimental settings and expected results may be simplified as long as they adequately support the motivation and methods**. Under these principles, manual correction was required for 22 samples, yielding an **extraction accuracy of 89.86%**.
>
> The two primary error types are: (1) safety-related papers triggering the LLM's refusal mechanism, which we resolved by adjusting prompts to enable responses; and (2) overly specific experimental designs and results that exceed what an idea requires, which we addressed by having the LLM perform secondary summarization to abstract these sections. Importantly, **paper Abstracts already provide excellent templates for idea representation**, so the Extraction Agent primarily refines and optimizes upon this foundation rather than constructing from scratch. **This makes our extraction pipeline highly reliable and fully capable of supporting downstream evaluation modules.**
>
> # W3&Q3. Conference outcome prediction vs. intrinsic idea quality
>
> Since the questions are consistent, please refer to our response to **Reviewer GWZN's W2&Q2**.
>
> To further demonstrate that InnoEval assesses **intrinsic idea quality rather than conference outcome prediction**, we have conducted additional experiments on non-existent ideas. Please refer to our response to **Reviewer KQBH's W3 (Testing on non-existing ideas)** for details.
>
> # W4&Q4. Prompt for Table 2
>
> Many thanks for your reminder. Due to space constraints, we provide the LLM-as-judge prompt for Table 2 via an **anonymous link**: **[URL](https://anonymous.4open.science/r/InnoEval-AC7E/rebuttal.pdf)**. To mitigate position bias, we randomly assign InnoEval and baseline reports to Report 1 or Report 2 positions during pairwise comparison. We will add this prompt in the Appendix during our revision.
>
> # W5&Q5. Leakage risk
>
> **We are fully aware that online search may introduce risks of data leakage and noise.** To mitigate these concerns, we have implemented the following safeguards:
>
> 1. **Source Restriction.** We strictly limit search scope to a curated set of popular blog and forum platforms (e.g., Medium, Reddit machine learning), deliberately excluding sites such as OpenReview that could potentially leak ground-truth decisions.
> 2. **Timestamp Filtering.** We explicitly filter out resources published after the paper's submission date, as these may contain human posterior knowledge that would compromise the integrity of evaluation.
> 3. **Content Filtering.** We filter resources that directly contain the original idea content to prevent information leakage and ensure fair assessment.
>
> These measures collectively ensure that web search enriches our knowledge base with diverse, timely perspectives while maintaining the rigor and fairness of the evaluation process.
>
>
>
> Thank you again for your constructive suggestions!
>
> **Please let us know if you have any further questions. If you find that our response addresses some of your concerns, would you kindly consider raising your rating score for our paper? We greatly appreciate your consideration.**
>
> Best Regards,
>
> Authors of InnoEval

---

> > ### Author Rebuttal · Reviewer_U5FS · 2026-04-01
> >
> > Thank you to the authors for the detailed rebuttal. The responses to W1–W4 and the corresponding questions are satisfactory and have addressed the majority of my original concerns. Regarding W5, while the described safeguards (source restriction, timestamp filtering, and content filtering) are reasonable, the leakage risk has not been fully resolved without empirical validation. That said, given the overall quality of the rebuttal, I am raising my score from 3 to 4.

---

> > > ### Author Response · Authors · 2026-04-01
> > >
> > > Dear Reviewer U5FS,
> > >
> > > Thank you very much for your recognition of our rebuttal and your reconsideration of our paper. Your review has been instrumental in substantially improving the quality of our manuscript.
> > >
> > > Regarding the web leakage concern, we conduct the n-gram similarity calculations between ideas and web content based on the retrieval results generated by InnoEval on our point-wise dataset. The results are as follows:
> > >
> > > | n    | >=0.01 | >=0.05 | >=0.10 | >=0.25 | >=0.50 |
> > > | ---- | -----: | -----: | -----: | -----: | -----: |
> > > | 2    |    192 |    160 |    149 |     25 |      0 |
> > > | 3    |    141 |     44 |     10 |      0 |      0 |
> > > | 4    |     61 |     14 |      2 |      0 |      0 |
> > > | 5    |     13 |      1 |      0 |      0 |      0 |
> > >
> > > **When n=5, the number of samples with similarity exceeding 0.10 is zero**, which can sufficiently demonstrate that the retrieved web content does not exhibit leakage issues. We hope this result can further addresses your concern.
> > >
> > > Thank you once again for your valuable time and constructive feedback.
> > >
> > > Best Regards,
> > >
> > > Authors of InnoEval

---

### Official Review · Reviewer_GWZN · 2026-03-11

**Soundness:** 3
**Presentation:** 4
**Significance:** 3
**Originality:** 3
**Overall Recommendation:** 4
**Confidence:** 3

**Summary:**

This paper proposes InnoEval, a framework for research idea evaluation that combines heterogeneous knowledge retrieval, fine-grained grounding, multi-dimensional scoring, and multi-perspective reviewer personas. It also introduces evaluation datasets for point-wise, pair-wise, and group-wise idea assessment, and reports improvements over several baselines on these tasks.

**Compliance With Llm Reviewing Policy:**

Affirmed.

**Final Justification:**

It's very clear that the paper is well polished, with sufficient evidence supporting the contributions. My concerns are resolved in general. As I'm not very confident with the topic, I therefore vote a weak acceptance. I appreciate the author's detailed rebuttal, and I also appreciate the author's own quality check instead of some random AI-written charts and content. I therefore increase the presentation.

**Key Questions For Authors:**

1. Which components are most responsible for the gains, beyond the current ablations?
2. How much does the benchmark reflect conference outcome prediction versus true intrinsic idea quality?

**Limitations:**

Yes, but the discussion could be strengthened, especially around benchmark assumptions, dependence on conference labels, and the gap between evaluation performance and true scientific judgment.

**Strengths And Weaknesses:**

I think this is a good paper overall. The problem is important and timely: idea generation has advanced quickly, while idea evaluation remains much less developed. The paper presents a relatively comprehensive framework that goes beyond simple LLM-as-a-judge evaluation by incorporating retrieval, grounding, multiple evaluation dimensions, and reviewer diversity. The empirical section is also fairly extensive, covering quantitative results, qualitative comparisons, human evaluation, and ablations. Overall, the work feels ambitious and practically useful.

My concerns are mostly about clarity and overclaiming rather than the overall direction. The system is quite large, so it is sometimes difficult to disentangle which components are most essential. Some conclusions, especially those relating the framework to “human cognition” or real scholarly consensus, feel a bit stronger than what the experiments fully establish. I also think the evaluation setup depends substantially on the paper-label formulation extracted from accepted/rejected submissions, so the benchmark may still be somewhat tied to conference outcomes rather than intrinsic research quality. In addition, the manuscript includes reviewer-facing instruction text beyond normal template boilerplate, which should be removed.

That said, I still view the paper positively. The benchmark and framework are both broad, the results are strong, and this seems like the kind of system paper that can be useful to future work even if some components would benefit from further refinement.

---

> ### Author Rebuttal · Authors · 2026-03-30
>
> Dear Reviewer GWZN,
>
> We are deeply grateful for your valuable time and insightful feedback. Below are our detailed responses to your concerns:
>
> # W1&Q1. Most responsible components for the gains
>
> We conducted additional ablation studies on the Extraction Agent and Search module, which were not included in the original paper.
>
> | |F1_2|F1_3|
> | ---------------- | -------------- | -------------- |
> |**InnoEval**|**75.74**|**74.56**|
> |w/o Extraction|69.71 (-6.03)|67.06 (7.50)|
> |w/o Search|53.46 (-22.28)|40.55 (-34.01)|
> |w/o Grounding|69.34 (-6.40)|68.64 (-5.92)|
> |w/o Personalized|67.71 (-8.03)|61.66 (-12.90)|
>
> |Compair|Rationality|Supportiveness|Depth|Constructiveness|Overall Quality|
> | ---------------- | ----------- | -------------- | ------ | ---------------- | --------------- |
> |w/ ScholarEval|Win(%)|Win(%)|Win(%)|Win(%)|Win(%)|
> |**InnoEval**|**67.28%**|**61.75%**|**70.51%**|**84.79%**|**71.89%**|
> |w/o Extraction|65.44%|58.06%|66.36%|80.18%|65.44%|
> |w/o Search|33.18%|23.04%|38.25%|31.80%|35.02%|
> |w/o Grounding|59.91%|53.46%|63.13%|74.65%|64.06%|
> |w/o Personalized|52.07%|50.23%|58.06%|70.51%|58.99%|
>
> The results clearly demonstrate that **the Search module is the most critical component**. This is because research ideas are inherently knowledge-intensive entities; without retrieved evidence as reference, the evaluation lacks grounding in relevant background, severely compromising its rationality, supportiveness, constructiveness, etc.
>
> **The Personalized evaluation mechanism ranks second in importance.** Scientific assessment fundamentally relies on consensus among diverse experts rather than single-authority judgment. Removing personalized personas reverts the system to vanilla LLM-as-a-Judge, which introduces greater bias and suffers from limited individual knowledge horizons and narrow perspectives, resulting in less comprehensive and thorough evaluations.
>
> These findings align with the core principles of InnoEval: knowledge-grounded reasoning and multi-perspective deliberation are both essential for faithful emulation of human-level scholarly review.
>
> # W2&Q2. Conference outcome prediction vs. intrinsic idea quality
>
> We would like to emphasize that predicting decision labels serves solely as an **objective proxy** to quantitatively evaluate our system's effectiveness, as obtaining gold-standard intrinsic quality scores from human experts at scale is practically infeasible. Our mapping from evaluation reports to labels strictly follows the assessment content generated by InnoEval—we merely align review comments with final decisions according to standard top-tier AI conference scoring rubrics, and this alignment procedure is applied consistently across all baselines.
>
> We also deeply recognize that **label prediction alone is insufficient to demonstrate true evaluation quality**. Therefore, we have conducted extensive complementary analyses to validate that InnoEval captures intrinsic research merit beyond superficial outcome prediction:
>
> 1. **Qualitative Evaluation.** Systematic comparison across five dimensions (Rationality, Supportiveness, Depth, Constructiveness, Overall Quality) using GPT-4o-mini (different from DeepSeek-V3.2 used in our main experiments) to avoid inherent bias.
> 2. **Human Evaluation.** Correlation analysis between InnoEval scores and real peer-review comments, plus direct expert scoring, demonstrating strong human alignment.
> 3. **Idea Optimization.** InnoEval's feedback integrated into ResearchAgent's pipeline significantly improves idea quality across problem formulation, methodology, and experimental design.
>
> To further demonstrate that InnoEval assesses **intrinsic idea quality rather than conference outcome prediction**, we have conducted additional experiments on non-existing ideas. Please refer to our response to **Reviewer KQBH's W3 (Testing on non-existing ideas)** for details.
>
> # W3. Instruction text beyond normal template
>
> Thank you for your careful observation. However, we would like to clarify that **this text was not added by us**. The additional instruction text is an official anti-LLM-reviewing measure injected **by the conference organizers**, not part of our manuscript submission. We have not modified the template beyond the standard required content.
>
>
>
> Thank you again for your constructive suggestions!
>
> **Please let us know if you have any further questions. If you find that our response addresses some of your concerns, would you kindly consider raising your rating score for our paper? We greatly appreciate your consideration.**
>
> Best Regards,
>
> Authors of InnoEval

---

> > ### Author Rebuttal · Reviewer_GWZN · 2026-04-04
> >
> > I appreciate the author's rebuttal. My concerns are resolved in general.

---

> > > ### Author Response · Authors · 2026-04-04
> > >
> > > Dear Reviewer GWZN,
> > >
> > > We once again express our sincere gratitude for your valuable time and effort. Your review comments have been immensely helpful in improving the quality of our paper.
> > >
> > > **We would like to cordially invite you to consider raising your rating score for our paper, given that you mentioned our rebuttal has fully addressed your concerns**. We have supplemented extensive experiments, including ablation studies on each module of our framework and non-existing idea evaluation. We believe our efforts are worthwhile. Please feel free to remind us if you have any further concerns, as your feedback is very important to us.
> > >
> > > Thank you once again for your dedication! We wish you all the best in your research and life.
> > >
> > > Best Regards,
> > >
> > > Authors of InnoEval

---

### Official Review · Reviewer_KQBH · 2026-03-13

**Soundness:** 3
**Presentation:** 4
**Significance:** 3
**Originality:** 2
**Overall Recommendation:** 4
**Confidence:** 3

**Summary:**

This work proposes an evaluation framework that is knowledge-grounded and multi-dimensional to judge an idea. The evaluation framework consists of multiple modules, including query generation, deep search, review board, and so on. As an output, it gives a diagnostic report that contains search results, revision recommendations, and so on.  The framework enables point-wise and pair/group-wise evaluation.

**Compliance With Llm Reviewing Policy:**

Affirmed.

**Final Justification:**

The authors' responses to my concerns about the difficulty of judging innovation itself and the potential issue of web search are resolved.

**Key Questions For Authors:**

I was curious why the authors were open to web opinions during the search. Some might argue that including web forums or blogs hurts, as they may contain incorrect information. What’s the authors’ take on this?

**Limitations:**

yes

**Strengths And Weaknesses:**

The study is well-structured and easy to follow, as the authors start with the problem definition, and then move on to each of the InnoEval components and corresponding experiments.

I am not quite sure that the two classification tasks, predicting if it is accepted or further receives Highlight, in the experiment settings indeed test how innovative the research ideas are, since this is more likely to be a research review on existing ideas. I wonder if the authors could test the framework with non-existing ones.

(+ I might miss, but wouldn’t the search part find out the answer anyway or get hinted from the web forums?)

---

> ### Author Rebuttal · Authors · 2026-03-30
>
> Dear Reviewer KQBH,
>
> We are deeply grateful for your valuable time and insightful feedback. Below are our detailed responses to your concerns:
>
> # W1. Predicting labels vs. real innovation evaluation
>
> We would like to emphasize that predicting decision labels serves solely as an objective proxy to quantitatively evaluate our system's effectiveness, as this is the only practical way to obtain gold-standard labels from human experts at scale. Our mapping from evaluation reports to labels strictly follows the assessment content generated by InnoEval. We merely align review comments with final decisions according to standard top-tier AI conference scoring rubrics, and this alignment procedure is applied consistently across all baselines.
>
> We also deeply recognize that label prediction alone is insufficient to demonstrate evaluation quality. Therefore, we have conducted extensive complementary analyses:
>
> 1. **Qualitative Evaluation.** Systematic comparison across five dimensions (Rationality, Supportiveness, Depth, Constructiveness, Overall Quality) using GPT-4o-mini (different from DeepSeek-V3.2 used in our main experiments) to avoid inherent bias.
> 2. **Human Evaluation.** Correlation analysis between InnoEval scores and real peer-review comments, plus direct expert scoring, demonstrating strong human alignment.
> 3. **Idea Optimization.** InnoEval's feedback integrated into ResearchAgent's pipeline significantly improves idea quality across problem formulation, methodology, and experimental design.
>
> Finally, we wish to underscore that **idea evaluation is holistic, not solely novelty**. A seemingly "groundbreaking" idea may sound highly novel, yet if it lacks feasibility, it cannot constitute a valuable research contribution. This multi-dimensional philosophy is precisely what distinguishes InnoEval from simplistic novelty-focused evaluation methods.
>
> # W2. The problem of using web search
>
> We believe that in contemporary academia, particularly in computer science, research papers are no longer the sole form of scholarly output. High-quality academic content also exists extensively in blogs, technical reports, and code repositories, all of which constitute critical components of the evolving research landscape. This "living knowledge" captures practical implementation insights, community discussions, and emerging trends that traditional literature often fails to encompass promptly.
>
> **We are fully aware that online search may introduce risks of data leakage and noise.** To mitigate these concerns, we have implemented the following safeguards:
>
> 1. **Source Restriction.** We strictly limit search scope to a curated set of popular blog and forum platforms (e.g., Medium, Reddit machine learning), deliberately excluding sites such as OpenReview that could potentially leak ground-truth decisions.
> 2. **Timestamp Filtering.** We explicitly filter out resources published after the paper's submission date, as these may contain human posterior knowledge that would compromise the integrity of evaluation.
> 3. **Content Filtering.** We filter resources that directly contain the original idea content to prevent information leakage and ensure fair assessment.
>
> These safeguards ensure diverse, timely perspectives without compromising evaluation rigor.
>
> # W3. Testing on non-existing ideas
>
> To further demonstrate InnoEval's effectiveness on non-existing ideas, we conducted additional experiments using 100 recently accepted high-scoring (>=6.5) papers from ICLR 2026 as anchor papers. We employed [GROBID](https://github.com/grobidOrg/grobid) to identify key citations and strictly followed ResearchAgent's protocol to generate research ideas via LLM, treating these as non-existing ideas. We further employed the Extraction Agent to normalize both the anchor idea and LLM-generated idea into a consistent format, ensuring no substantial disparities beyond semantic content. Qualitative LLM-as-Judge and quantitative comparisons (anchor papers as positive, generated ideas as negative) demonstrate InnoEval's superior evaluation quality and prediction accuracy versus all baselines:
>
> | |Comparison|Rationality|Supportiveness|Depth|Constructiveness|Overall Quality|
> | ---- | ---- | ---- | ---- | ---- | ---- | ---- |
> | |Acc.|Win / Loss|Win / Loss|Win / Loss|Win / Loss|Win / Loss|
> |CoT|64|92% / 4%|97% / 2%|89% / 5%|91% / 4%|91% / 6%|
> |RAG|68|90% / 3%|89% / 11%|85% / 10%|90% / 8%|88% / 9%|
> |ResearchAgent|76|88% / 7%|82% / 9%|82% / 14%|85% / 11%|86% / 11%|
> |InternAgent|85|85% / 11%|84% / 7%|84% / 7%|82% / 9%|81% / 13%|
> |ScholarEval|89|73% / 21%|66% / 28%|69% / 30%|80% / 16%|70% / 23%|
> |InnoEval|98|———|———|———|———|———|
>
>
>
> Thank you again for your constructive suggestions!
>
> **Please let us know if you have any further questions. If you find that our response addresses some of your concerns, would you kindly consider raising your rating score for our paper? We greatly appreciate your consideration.**
>
> Best Regards,
>
> Authors of InnoEval

---

> > ### Author Rebuttal · Reviewer_KQBH · 2026-04-06
> >
> > Thanks for the response. I increase my score accordingly

---

> > > ### Author Response · Authors · 2026-04-06
> > >
> > > Dear Reviewer KQBH,
> > >
> > > Thank you very much for your recognition of our rebuttal and your reconsideration of our paper. Your review has been instrumental in substantially improving the quality of our manuscript.
> > >
> > > Best Regards,
> > >
> > > Authors of InnoEval

---

### Decision · Program_Chairs · 2026-04-30

**Decision:**

Accept (regular)

**Comment:**

The paper introduces InnoEval, a framework for research idea evaluation that combines a heterogeneous deep knowledge retrieval engine with a multi-agent review board of diverse academic personas scoring along five dimensions. The paper build benchmarks from ICLR 2025 and NeurIPS 2025 submissions and show gains over RAG and agent baselines on point-wise, pair-wise, and group-wise tasks.

### Strengths
- The motivation is clear and the identified gaps (grounding, deliberation, multi-criteria) are well-articulated.
- The problem is timely given the widening gap between idea generation and evaluation.
- The framework goes beyond naive LLM-as-a-judge via retrieval grounding, multi-dimensional scoring, and reviewer diversity.
- Empirical gains are demonstrated across three evaluation regimes (point/pair/group-wise) with alignment to human judgments.

### Weaknesses
- Some reviewers questioned whether the classification tasks (predicting acceptance or Highlight status) test innovation, since they test on existing published ideas than novel ones.
- Data leakage via web retrieval: multiple reviewers flagged that retrieving from blogs, forums, and code repositories could expose acceptance-correlated signals or even the source paper itself, potentially inflating reported gains.
- Quality of heterogeneous online proxies: one review noted that signals such as citations, venue, stars, and popularity do not guarantee factual correctness or robustness to noisy/manipulated web content, with limited failure-mode analysis.
- Conference labels as a noisy proxy for intrinsic idea quality: several reviewers argued that acceptance decisions conflate writing quality, experimental completeness, and track effects with idea quality, making it unclear whether the system measures intrinsic merit or acceptance-correlated signals.
- one review felt the framing around emulating human-level judgment is stronger than what the experiments establish.
- Missing ablations and cost analysis: reviewers asked for systematic curves over persona count, retrieval rounds, and retained items, plus token usage, API cost, and latency comparisons against baselines.

The rebuttal addressed several of the above concerns. Leakage risks were countered with source restrictions (excluding OpenReview), timestamp filtering, content filtering, and follow-up n-gram overlap statistics. Additional ablations isolated the Search module and Personalized evaluation as the largest contributors, and cost/latency tables plus recommended default configurations were supplied.
The remaining open issues are about the general scope and framing and the general impression of the work.